# Image Enhancement: A Necessity for Effective Underwater Object Detection?

## Abstract

Underwater vision is essential for applications such as marine engineering, aquatic robotics, and environmental monitoring. However, severe image degradation caused by light absorption and scattering often compromises object detection performance. Although underwater image enhancement (UIE) intuitively seems beneficial for restoring visual information and improving detection accuracy, its actual impact remains unclear. This work systematically evaluates state-of-the-art enhancement models and investigates their effects on underwater object detection to answer the key question: **"Is UIE necessary for accurate object detection?"** We conducted a systematic evaluation of 20 representative UIE algorithms—spanning traditional methods, convolutional neural networks (CNNs), generative adversarial networks (GANs), Transformers, and Diffusion models. These methods are applied to two benchmark datasets, RUOD and URPC2020, producing 21 domain variants per dataset (raw + 20 enhanced). To rigorously assess the effect of enhancement on detection, we trained five object detectors on each domain, resulting in 210 unique model configurations (5 detectors × 21 domains × 2 datasets). Our findings reveal that, contrary to intuitive expectations, most enhancement techniques actually **degrade** detection accuracy. Only well-designed methods, such as diffusion-based approaches that preserve key low-level features without introducing artificial distortions, can minimize this negative impact. These results provide critical insights into the role of enhancement in underwater vision and highlight important considerations for future research.

## 1 Introduction

In recent years, underwater vision has emerged as a crucial area of research due to its wide-ranging applications in marine engineering, environmental monitoring, and the maintenance of underwater infrastructure (Anwar & Li, 2020; Cui et al., 2020). A fundamental challenge in these applications is object detection (OD), which enables the accurate identification and localization of underwater objects. However, detection performance is often compromised due to the severe degradation of underwater images. Factors such as light absorption, scattering, and backscattering lead to low contrast, color distortion, and blurring, making it difficult for deep learning models to reliably detect objects in underwater environments.

A natural solution to this problem is UIE, which aims to restore lost visual information and improve image quality. Intuitively, enhanced images should improve OD accuracy. However, the actual impact of enhancement on detection performance remains unclear. While reserchers suggest that enhancement aids detection, studies in other domains (*e.g.,* automatic speech recognition ) show that signal restoration can sometimes degrade recognition accuracy (Iwamoto et al., 2022; Menne et al., 2019), indicating that perceptual quality does not always align with recognition accuracy. A systematic evaluation of enhancement techniques for underwater OD remains lacking, leaving a gap in understanding their real-world effectiveness.

This paper addresses the key question: **"Does underwater image enhancement improve object detection performance?"** To investigate this, we conducted a large-scale empirical study analyzing the correlation between enhancement and detection performance. We selected 20 representative underwater image enhancement algorithms, encompassing traditional methods (Hsu & Cheng, 2021; Fu et al., 2014a; Hummel, 1977; Jr et al., 2013) and SOTA deep learning approaches including CNNs (Li et al., 2020a;b; Wang et al., 2021; Huo et al., 2021) , GANs (Li et al., 2017a; Fabbri et al., 2018a; Wang et al., 2019; Desai et al., 2022; Wang et al., 2023; Jiang et al., 2022) , Transformers (Tang et al., 2022; Khan et al., 2024; Wang et al., 2024) , and Diffusion models (Zhao et al., 2024a; Du et al., 2025; Tang et al., 2023) . These algorithms were individually applied to two benchmark underwater datasets, RUOD (FU2, 2023) and URPC2020 (Liu et al.,

2021), followed by comprehensive qualitative and quantitative analyses of the enhanced images. Following this, we independently trained and tested five widely used OD models on both raw and enhanced image sets—a total of 210 models, including 200 trained on enhanced images and 10 on raw underwater images. This extensive study provides a thorough evaluation of how UIE techniques influence OD performance.

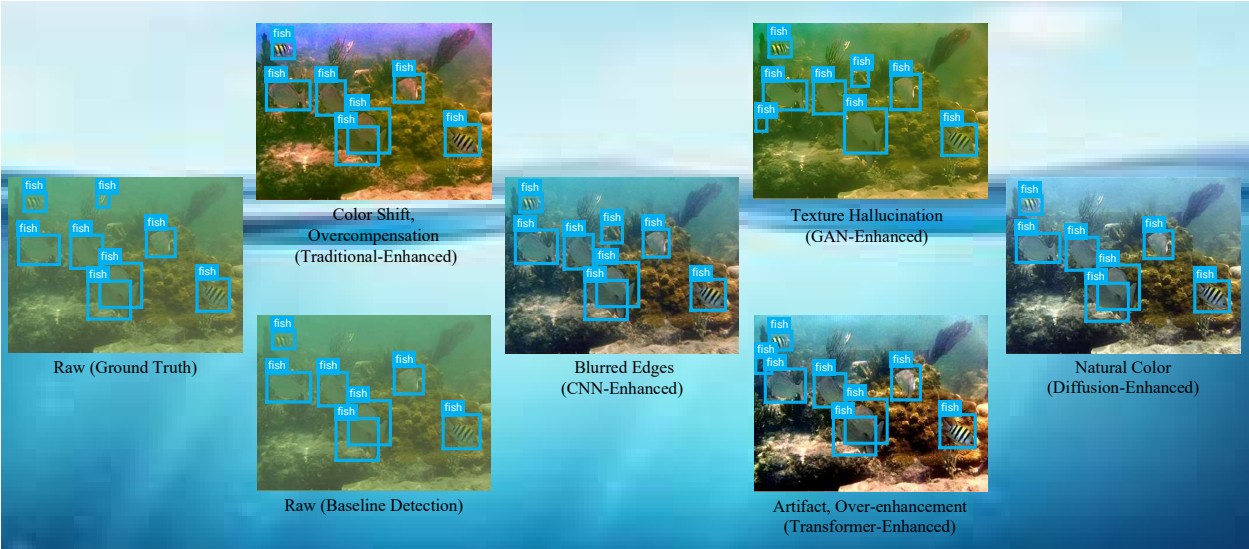

Figure 1: Qualitative comparison of UIE methods and their effects on OD. The raw image and its detection result serve as a baseline. Each enhanced version—Traditional WB (Hsu & Cheng, 2021), CNN-based PRWNet (Huo et al., 2021), GAN-based WaterGAN (Li et al., 2017a), Transformer-based AutoEnhancer (Tang et al., 2022), and Diffusion-based UIEDP (Du et al., 2025)—introduces distinct visual changes, illustrating how UIE alters image statistics and influences detection performance.

Our empirical observations revealed several key findings. Contrary to intuition, most enhancement techniques actually reduced OD performance. Although visually improved, enhanced images often introduced artifacts and domain shifts that impaired detection (Fig. 1). Preserving edge details is crucial for detecion accuracy, while unintended color shifts and additional noise degrade performance. Although contrast changes have little effect, color richness and saturation play an important role in supporting accurate detection. Effective enhancement methods must preserve structural integrity, maintain color consistency, and minimize noise and artifacts. Among all evaluated approaches, diffusion-based methods, such as WF-Diff (Zhao et al., 2024a) and UIEDP (Du et al., 2025), demonstrate clear superiority in balancing color correction and detail preservation, consistently achieving higher detection accuracy across multiple detectors. Building on these findings, the main contributions of this paper are as follows:

- We present a comprehensive study assessing the impact of UIE on OD performance, providing ta large-scale empirical evaluation of enhancement techniques.
- Through extensive experiments and analysis, we identify critical limitations in existing UIE algorithms, particularly their lack of robustness and adaptability to complex underwater environments.
- We reveal that current underwater image quality assessment metrics do not correlate well with OD performance, highlighting the urgent need for task-aware UIE evaluation criteria that better reflect downstream task utility.

This study does not propose new image enhancement techniques but instead evaluates whether preprocessing with enhancement methods improves object detection in underwater environments. We aim to inspire further research into integrated approaches that effectively combine image enhancement with OD, ultimately enhancing both visual quality and detection accuracy in underwater vision systems.

## 2 Related Work

**Underwater Image Degradation and Enhancement**   Underwater images suffer from light absorption and scattering, causing color distortion, low contrast, and haze. According to the underwater image formation model, $I(x,y) = J(x,y)e^{-\beta d(x,y)} + B(1 - e^{-\beta d(x,y)})$, where $J(x,y)$ is the true radiance, $\beta$ the attenuation

coefficient, $d(x, y)$ the object distance, and $B$ the background light. UIE methods aim to recover $J(x, y)$ from $I(x, y)$ to restore visual clarity and support tasks such as object detection. Traditional UIE methods adjust intensity distributions and pixel values to improve contrast and color balance. Color-correction (Finlayson & Trezzi, 2004) and Retinex-based (Fu et al., 2014b) approaches refine illumination and local contrast, while prior-based methods remove haze using scene priors. Contrast enhancement techniques (Bai et al., 2020) further improve visibility and edge sharpness. With the rise of deep learning, CNN-based (Zhai et al., 2023; Li et al., 2020a;b) and GAN-based (Li et al., 2017b; Fabbri et al., 2018b; Zhai et al., 2022) methods have become dominant for UIE. Transformer-based models (Chen et al., 2021; Wang et al., 2022; Dosovitskiy et al., 2020) further extend these capabilities through self-attention, while diffusion models (Ho et al., 2020; Song et al., 2021) provide more stable optimization and stronger restoration performance than GANs (Goodfellow et al., 2014) and VAEs (Higgins et al., 2016; Kingma & Welling, 2013). Recent studies also integrate UIE with object detection, and frequency-domain approaches such as WF-Diff (Zhao et al., 2024b) leverage wavelet–Fourier interactions to refine both high- and low-frequency features, achieving state-of-the-art results.

**Object Detection.** OD identifies and localizes objects by predicting category labels and bounding boxes. Modern OD methods are generally categorized as two-stage or one-stage approaches. Two-stage detectors (**e.g.**, R-CNN, Fast R-CNN, and Faster R-CNN (Girshick et al., 2014; Ren et al., 2015)) use a proposal–refinement pipeline for high accuracy but slower inference. One-stage detectors, such as SSD (Liu et al., 2016), RetinaNet (Lin et al., 2020), and YOLO-NAS (AI, 2023), perform detection in a single pass for real-time efficiency. Both paradigms continue to evolve with transformer-based and anchor-free designs that further push OD performance.

**Joint Task.** Prior work has examined whether low-level enhancement benefits high-level vision tasks such as OD. Early studies (Pei et al., 2018; 2021) showed that enhancement yields limited or inconsistent OD gains and may even degrade performance. Similar observations were reported for other restoration tasks: dehazing does not improve semantic segmentation (Hahner et al., 2019), and de-raining can worsen OD accuracy (Li et al., 2019). For underwater vision, results remain mixed. Some studies reported case-specific improvements for segmentation and saliency detection (Zhuang et al., 2022), whereas others found that most UIE methods hinder OD performance (Chen et al., 2020; Liu et al., 2022b;a). These inconsistencies highlight that the interaction between enhancement and detection is still poorly understood. Recent efforts attempt end-to-end designs that jointly optimize UIE and OD (Liu et al., 2022a; Jiang et al., 2021), yet their benefits remain inconclusive. This motivates our study: a systematic evaluation across diverse UIE algorithms and detectors to clarify when, how, and why enhancement helps or harms underwater OD.

# 3 Systematic Analysis of UIE-Induced Changes and Their Impact on Detection

Before investigating whether and how UIE contributes to OD performance, we first investigate its direct effects on image quality itself. Specifically, we examine how UIE influences key visual and structural aspects such as color consistency, edge and texture preservation, and artifact introduction.

We begin by outlining the experimental setup, including the selected datasets and UIE methods. We then present an analysis of the UIE results using both qualitative (visual inspection) and quantitative (reference-free image quality metrics) evaluations. First, we describe the experimental setup on UIE. Next, we analyze the pre-processing results from various UIE algorithms using both qualitative and quantitative methods.

**Selected Underwater Image Dataset**. We conducted our experiments on two underwater datasets: (1) RUOD (FU2, 2023), the Real-World Underwater OD dataset, contains 14,000 high-resolution murky underwater images, with 9,800 images used for training and approximately 75,000 annotations spanning 10 categories of aquatic objects such as divers, plants, and various marine animals; and (2) URPC2020 (Liu et al., 2021), the Underwater Robot Professional Contest dataset containing both real and artificial underwater environments, consists of 5,543 underwater images categorized into four classes: holothurian, echinus, scallop, and starfish. For our study, we randomly split the dataset into 4,100 training images and 1,443 testing images. The use of both datasets helps us to draw more generalized conclusions.

**Selected UIE Algorithms.** To comprehensively investigate the impact of UIE on underwater OD, we explore both traditional and learning-based UIE methods. Traditional UIE techniques enhance obscured details by adjusting intensity distribution or applying pixel-level transformations to improve contrast. We select representative approaches, including color constancy methods (White Balance-based WB(Hsu & Cheng,

2021), Retinex (Fu et al., 2014a)), contrast enhancement techniques (Histogram Equalization (HE) (Hummel, 1977)), prior knowledge-based methods (UDCP (Jr et al., 2013)), and transform-domain-based approaches (wavelet transform-based PRWNet (Huo et al., 2021)). We also studied top-performing deep learning-based UIE methods leveraging architecture: CNN-based (UWCNN (Li et al., 2020a), Water-Net (Li et al., 2020b), and UIEC$^2$-Net (Wang et al., 2021)), GAN-based (WaterGAN (Li et al., 2017a), UGAN (Fabbri et al., 2018a), UWGAN (Wang et al., 2019), AquaGAN (Desai et al., 2022), TUDA (Wang et al., 2023), and TOPAL (Jiang et al., 2022)), Transformer-based (AutoEnhancer (Tang et al., 2022), Spectroformer (Khan et al., 2024), and UIE-Convformer (Wang et al., 2024)), and Diffusion-based (WF-Diff (Zhao et al., 2024a), UIEDP (Du et al., 2025), and DM_Underwater (Tang et al., 2023)). To ensure a fair and unbiased comparison of UIE models, we re-trained these models on the UIEB dataset (Li et al., 2020b) and the EUVP dataset (Islam et al., 2020) to generate the corresponding enhanced results.

## 3.1 Qualitative Analysis of Enhanced Results

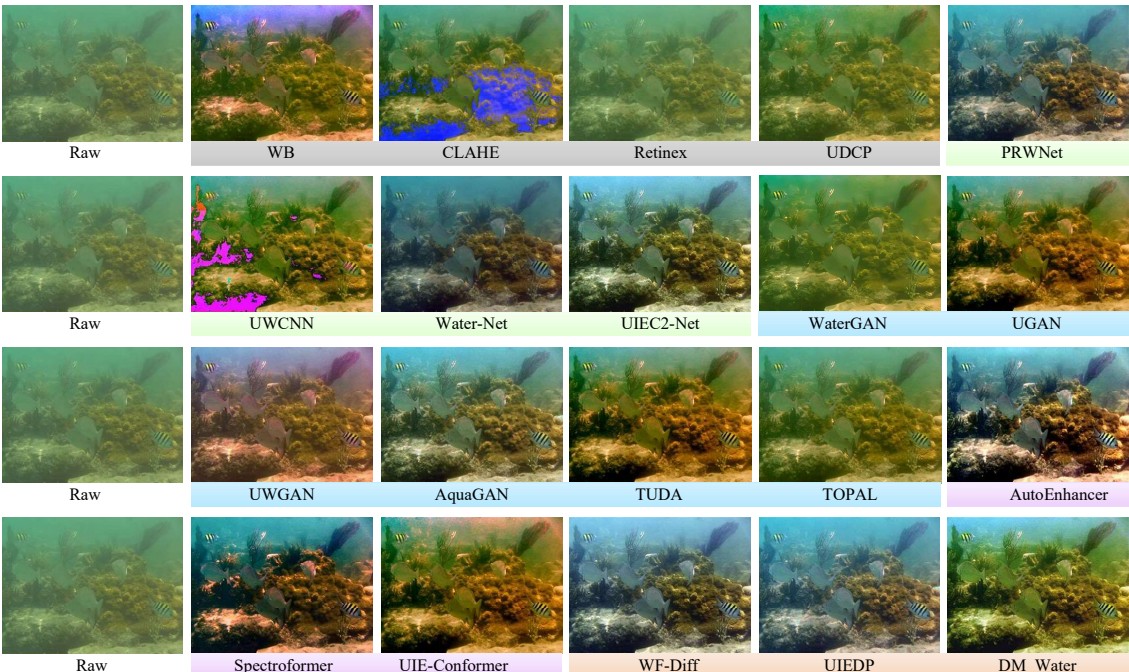

Figure 2: Visual comparison of UIE results. Raw images are shown for reference. Methods are color-coded: Traditional ; CNNs ; GANs ; Transformers ; and Diffusions .

**Qualitative Evaluation.** The results of different UIE are shown in Fig. 2. Due to light absorption in water, red light disappears first, followed by green and blue, leading to color distortions, reduced contrast, and poor visibility. This selective attenuation of light wavelengths results in greenish or bluish underwater images, as observed in the raw images in Fig. 2(a). Color deviation significantly impacts the visual quality of underwater images, making it difficult to discern objects and fine details accurately.

**(1) Traditional Methods.** White Balance (WB) (Hsu & Cheng, 2021) and Retinex-based approaches (Fu et al., 2014a) partially correct global color shifts but often introduce overcompensation in certain areas, leading to unnatural yellowish or purplish tones. In particular, WB amplifies local illumination inconsistency, while Retinex enhances global brightness at the expense of suppressing local contrast. CLAHE (Hummel, 1977) improves local contrast but generates visible artifacts in homogeneous regions, resulting in artificial texture patterns. UDCP (Jr et al., 2013), while effective in haze removal, introduces a reddish color bias that disrupts the scene's original color composition.

**(2) CNN-Based Deep Learning Methods.** PRWNet (Huo et al., 2021) shows notable improvement in preserving fine-grained textures and structural edges through frequency-domain processing, although slight oversharpening is observed under heavy scattering conditions. UWCNN (Li et al., 2020a), Water-Net (Li et al., 2020b), and UIEC2-Net (Wang et al., 2021) enhance color vibrancy and contrast, yet they occasionally

over-saturate highlights, causing loss of fine detail in brightly lit regions. Some subtle halo effects are also visible around object boundaries.

**(3) GAN-Based Methods.** GAN-based models such as UGAN (Fabbri et al., 2018a) and AquaGAN (Desai et al., 2022) exhibit strong color correction capabilities but often introduce blurred artifacts and inconsistent texture reconstruction, especially at object contours and in highly degraded regions. These artifacts manifest as "smearing" effects, weakening object distinction.

**(4) Transformer-Based Methods.** AutoEnhancer (Tang et al., 2022), Spectroformer (Khan et al., 2024), and UIE-Convformer (Wang et al., 2024) produce smoother contrast gradients and more globally coherent color restoration compared to CNN and GAN models. They effectively suppress halo artifacts and better preserve structural boundaries, although occasional mild over-smoothing can reduce textural richness in detailed regions. **(5) Diffusion-Based Methods.** Diffusion models, notably WF-Diff (Zhao et al., 2024a) and UIEDP (Du et al., 2025), deliver the most balanced performance: restoring natural color tones, enhancing perceptual contrast without exaggeration, and maintaining edge sharpness. Notably, WF-Diff reduces noise amplification and minimizes color bleeding, leading to visually more stable and naturalistic results. DM_Water (Tang et al., 2023), a lightweight diffusion model, offers moderate enhancement but exhibits slight under-correction in highly turbid scenarios.

**Key Visual Insights: (1) Color Consistency:** Diffusion-based (Tang et al., 2023; Du et al., 2025; Zhao et al., 2024a) and Transformer-based (Khan et al., 2024; Wang et al., 2024) methods provide the most consistent color correction without introducing severe domain shifts. GAN-based methods (Fabbri et al., 2018a; Desai et al., 2022) often generate unnatural hues and inconsistent color patches.

**(2) Edge and Texture Preservation:** Frequency-domain models (*e.g.*, PRWNet, WF-Diff (Zhao et al., 2024a) ) and Transformer-based models (Tang et al., 2022; Khan et al., 2024; Wang et al., 2024)outperform others in preserving object edges and fine textures critical for visual perception and downstream OD tasks.

**(3) Artifact Introduction:** Traditional methods (*e.g.*, CLAHE (Hummel, 1977) and Retinex (Fu et al., 2014a)) and GAN-based methods (Fabbri et al., 2018a; Li et al., 2017b) often cause visible artifacts, including texture noise, color bleeding, and blurring, which reduce image realism.

**(4) Over-Saturation:** CNN-based methods (*e.g.*, UWCNN (Li et al., 2020a) and Water-Net (Li et al., 2020b)) tend to over-saturate bright regions, leading to color overshooting and highlight clipping. **Overall Observations:** Diffusion models, especially WF-Diff (Zhao et al., 2024a) and UIEDP (Du et al., 2025), achieve the best qualitative results, effectively balancing enhancement strength and naturalness. Transformer-based models closely follow, offering strong perceptual improvements with minimal artifact introduction. Traditional and GAN-based methods, while enhancing certain aspects, frequently compromise structural integrity and color fidelity.

## 3.2 Perceptual Quality Evaluation of UIE

To validate the visual findings, we employ widely used reference-free image quality assesment (IQA) metrics that enable objective and quantitative comparison across UIE methods.

**Perceptual Quality Metrics.** For UIE, since no ground truth (clear) images are available for raw inputs, we adopt five widely used reference-free IQA metrics: Average Gradient (AG) (Zhang et al., 2019b), Edge Intensity (EI) (Azmi et al., 2019), Information Entropy (IE) (Zhang et al., 2019a), Underwater Image Quality Measure (UIQM) (Yang & Sowmya, 2015), and Underwater Image Contrast Measure (Yang & Sowmya, 2015). AG measures sharpness and texture by calculating intensity gradients. EI quantifies edge strength, with higher values indicating more texture detail. IE reflects the amount of information in an image. UIQM is a composite score based on colorfulness (UICM), sharpness (UISM), and contrast (UIConM). Higher UIQM scores indicate better overall visual quality.

**Quantitative Comparisons.** The images from the selected datasets were enhanced using selected enhancement models and evaluated using the selected enhancement metrics. Table 1 summarizes the results. Below we discuss the performance of each category of enhancement method. (1) CNN models — Sharpness gains but structural instability. PRWNet achieves the highest AG on both datasets, indicating strong texture preservation. Water-Net, however, shows notably lower AG and EI especially on RUOD, implying noise amplification and structural distortion. Overall, CNNs output enhance sharpness but are inconsistent across

Table 1: UIE scores in AG (Zhang et al., 2019b), EI (Azmi et al., 2019), IE (Zhang et al., 2019a), UIQM (Yang & Sowmya, 2015), and UIConM (Yang & Sowmya, 2015) on **RUOD** (FU2, 2023) and **URPC2020** (Liu et al., 2021). Best in **bold**, second-best underlined. Methods are color-coded: Traditional , CNNs , GANs , Transformers , and Diffusions . Trad. – Traditional, Trans. – Transformer, Diff. – Diffusion.

| Methods | | RUOD (FU2, 2023) Dataset | | | | | URPC2020 (Liu et al., 2021) Dataset | | | | |
|---|---|---|---|---|---|---|---|---|---|---|---|
| Name | Type | AG ↑ | EI ↑ | IE ↑ | UIQM ↑ | UIConM ↑ | AG ↑ | EI ↑ | IE ↑ | UIQM ↑ | UIConM ↑ |
| RAW | - | 1.31 | 12.79 | 6.53 | 1.59 | 0.41 | 2.97 | 15.46 | 6.32 | 1.87 | 0.47 |
| WB (Hsu & Cheng, 2021) | Trad. | 1.29 | 13.63 | 6.18 | 1.70 | 0.43 | 1.15 | 10.14 | 7.08 | 1.53 | 0.35 |
| CLAHE (Hummel, 1977) | Trad. | 4.01 | 48.95 | 5.97 | 2.49 | 0.38 | 3.33 | 35.10 | 6.44 | 3.04 | 0.33 |
| Retinex (Fu et al., 2014a) | Trad. | 3.56 | 37.73 | 7.09 | 3.14 | 0.49 | 2.58 | 26.44 | 6.40 | 2.51 | 0.39 |
| UDCP (Jr et al., 2013) | Trad. | 2.69 | 28.62 | 7.31 | 2.65 | 0.54 | 3.48 | 24.47 | 6.69 | 2.30 | 0.70 |
| PRWNet (Huo et al., 2021) | CNN | **4.98** | 50.18 | 7.06 | 3.15 | 0.57 | **6.41** | 36.88 | 6.07 | 3.60 | 0.48 |
| UWCNN (Li et al., 2020a) | CNN | 3.81 | 31.59 | 7.13 | 3.34 | 0.78 | 3.21 | 36.38 | 6.69 | 2.78 | 0.55 |
| Water-Net (Li et al., 2020b) | CNN | 1.79 | 16.52 | 7.65 | 1.95 | 0.42 | 5.35 | 12.52 | **7.81** | 3.29 | 0.37 |
| UIEC²-Net (Wang et al., 2021) | CNN | 2.19 | 27.03 | 7.36 | 1.86 | 0.49 | 4.60 | 32.53 | 6.52 | 2.08 | 0.36 |
| WaterGAN (Li et al., 2017a) | GAN | 1.70 | 18.96 | 7.62 | 2.75 | 0.47 | 5.44 | 16.93 | 6.96 | 2.02 | 0.36 |
| UGAN (Fabbri et al., 2018a) | GAN | 2.09 | 48.65 | 7.04 | 3.47 | 0.54 | 2.49 | 39.42 | 6.02 | 3.73 | 0.66 |
| UWGAN (Wang et al., 2019) | GAN | 1.91 | 27.96 | 7.40 | 1.94 | 0.59 | 3.55 | 33.23 | 6.48 | 2.30 | 0.75 |
| AquaGAN (Desai et al., 2022) | GAN | 2.72 | 32.41 | **7.85** | 3.69 | 0.47 | 3.50 | 37.01 | 6.36 | **4.60** | 0.38 |
| TUDA (Wang et al., 2023) | GAN | 4.05 | 43.56 | 7.53 | 3.04 | 0.46 | 3.63 | 37.94 | 7.54 | 4.07 | 0.40 |
| TOPAL (Jiang et al., 2022) | GAN | 1.69 | 29.76 | 6.95 | 2.11 | 0.42 | 3.35 | 48.35 | 6.03 | 2.49 | 0.50 |
| AutoEnhancer (Tang et al., 2022) | Trans. | 1.73 | 38.96 | 6.98 | 1.63 | 0.54 | 5.37 | **56.85** | 7.49 | 3.89 | 0.64 |
| Spectroformer (Khan et al., 2024) | Trans. | 4.62 | 18.96 | 7.02 | 2.29 | 0.59 | 3.36 | 14.63 | 6.24 | 2.68 | 0.55 |
| UIE-Convformer (Wang et al., 2024) | Trans. | 2.08 | 42.18 | 7.39 | 3.35 | 0.41 | 2.51 | 34.09 | 6.14 | 2.80 | 0.31 |
| WF-Diff (Zhao et al., 2024a) | Diff. | 3.17 | **53.38** | 7.79 | **3.80** | **0.79** | 5.50 | 38.28 | 6.57 | 4.04 | **0.77** |
| UIEDP (Du et al., 2025) | Diff. | 3.22 | 32.64 | 6.90 | 3.13 | 0.56 | 3.65 | 25.70 | 6.33 | 3.53 | 0.40 |
| DM_Water (Tang et al., 2023) | Diff. | 4.19 | 48.96 | 7.02 | 3.48 | 0.53 | 4.14 | 42.28 | 6.11 | 4.29 | 0.44 |

scenes. (2) GAN methods — Strong color boost but lowest sharpness. AquaGAN achieves high UIQM scores, indicating strong improvements in colorfulness and contrast. However, the GAN category has the lowest average AG (2.36) on the RUOD dataset, suggesting weakened sharpness and fine-texture preservation. (3) Transformer methods — Stable globalcontrast enhancement TUDA and TOPAL obtain the highest EI on URPC2020, demonstrating effective global contrast improvement and edge preservation through attention-based modeling. (4) Diffusion methods — Superior overall performance. Diffusion-based approaches show the most consistent behavior across all metrics, with WF-Diff demonstrating superior performance, particularly in UIQM and UIConM on both datasets. This indicates its ability to enhance color vibrancy, improve contrast, refine soft edges, and effectively reduce fog effects. (5) Traditional techniques yield moderate enhancements but lag behind deep models in sharpness, texture richness. (6) Overall Trend — Deep models lead, metrics still limited. Deep learning approaches generally outperform traditional methods in perceptual metric space. However, consistent with exsiting findings (Liu et al., 2020), numerical quality scores do not always align with human visual perception. **This highlights the limitations of current underwater IQA metrics and underscores the need for perceptually aligned evaluation criteria**.

### 3.3 Distribution Shifts Introduced by UIE

**Distribution Shift Metrics.** While Section 3.2 evaluates perceptual quality, those metrics do not reveal how UIE alters the pixel, color, and texture statistics that form the basis of detector feature learning. To better characterize such effects, we quantify the distribution shift using RGB L1 intensity shift (Zhao et al., 2017), CIEDE2000 $\Delta E$ color drift (Sharma et al., 2005), RGB histogram distance (Swain & Ballard, 1991), Sobel-based edge shift (Sobel, 1968), VGG Gram-distance texture change (Gatys et al., 2016), and LPIPS perceptual deviation (Zhang et al., 2018). These measurements provide a distribution-level view of UIE behavior, which we later examine in relation to detection performance in Section 4.

**Results and Discussion.** Across all metrics, traditional UIE methods introduce only moderate distribution shifts, CNN-based approaches produce larger deviations, and GAN-based models exhibit the most

pronounced pixel-, color-, and texture-level changes. Diffusion-based enhancement is the most stable, remaining closest to the RAW domain and showing the weakest overall shift. These quantitative patterns mirror the downstream detection behavior reported in Section 4: UIE methods that induce stronger distribution shifts tend to suffer greater mAP degradation (Table 3). This provides empirical support for the observation that visually improved enhancement does not necessarily benefit object detection.

### 3.4 Stratified Sensitivity Across Scene Factors

We analyze UIE performance across object scale, water turbidity, and texture richness. These factors are derived from RUOD/URPC2020 metadata. Object scale follows COCO-style thresholds: small ($<32^2$ px), medium ($32^2$–$96^2$ px), and large ($>96^2$ px). Turbidity is defined using a dark-channel-based haze score, with images split at the dataset median into low and high turbidity. Texture richness is measured by average gradient magnitude and is divided at the median into low- and high-texture categories.

**Results and Discussion.** Table 4 in Appendix shows that small objects, high-turbidity scenes, and high-texture regions are most sensitive to UIE, explaining the uneven degradation reported in the main text. Small objects suffer the largest degradation under UIE (3.1 mAP). High-turbidity scenes show the strongest sensitivity to enhancement-induced color shift (4.2 mAP). Texture-rich regions amplify GAN-induced hallucinations (5.0 mAP). These trends quantitatively support our statements regarding edge preservation, color distortion, and texture artifacts.

### 3.5 UIE-Induced Attention Distortion

To understand why diffusion-based UIE better preserves detection performance while CNN-, GAN-, and Transformer-based methods often do not, we analyze how enhancement affects detector attention using both qualitative visualization and a quantitative alignment metric.

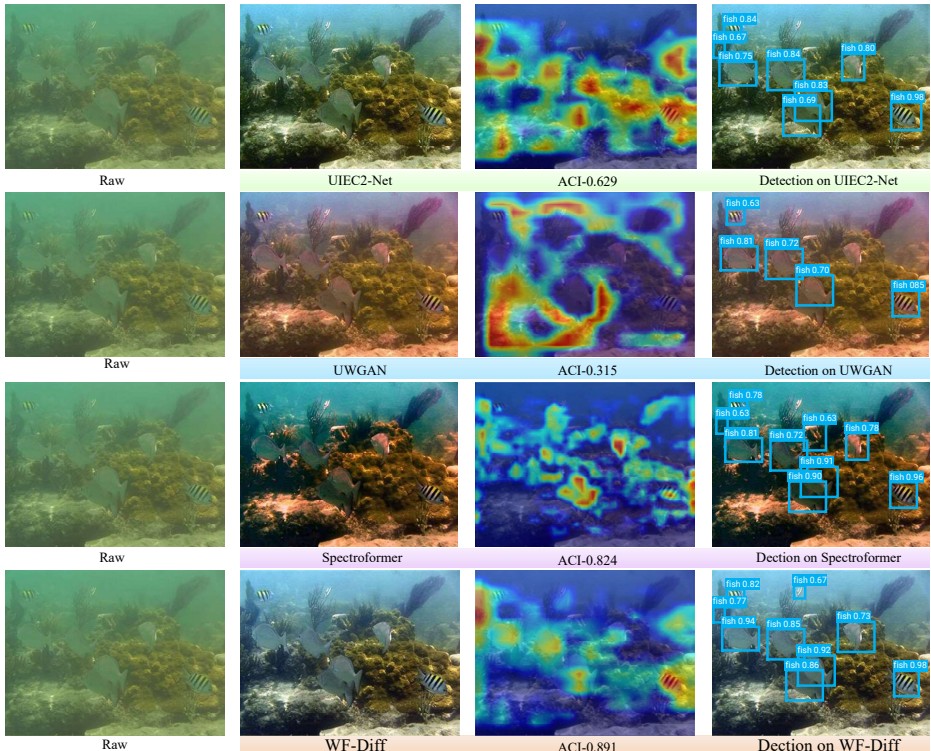

Figure 3: EigenCAM visualizations showing detector attention alignment across UIE methods. Diffusion-based models maintain focused activations near target boundaries, while CNN and Transformer methods exhibit dispersed or background-biased attention.

**EigenCAM:** We apply EigenCAM (Muhammad & Yeasin, 2020) to YOLO-NAS feature maps to visualize how UIE affects detector focus (Fig. 3), revealing whether enhancement preserves object-relevant structures or introduces distracting background activations. **Attention Concentration Index (ACI):** To measure

Table 2: **Baseline Pipeline: Train and Test in the Same Domain.** mAP$_{50:95}$) of Faster R-CNN, Cascade R-CNN, RetinaNet, SSD, and YOLO-NAS. Best in **bold**, second-best underlined. Trad. – Traditional, Trans. – Transformer, Diff. – Diffusion, C-RCNN – Cascade R-CNN.

| Methods | | RUOD (FU2, 2023) Dataset | | | | | URPC2020 (Liu et al., 2021) Dataset | | | | |
|---|---|---|---|---|---|---|---|---|---|---|---|
| Name | Type | Faster R-CNN | C-RCNN | RetinaNet | SSD | YOLO-NAS | Faster R-CNN | C-RCNN | RetinaNet | SSD | YOLO-NAS |
| RAW | - | **57.91** | 59.44 | **50.71** | **45.96** | **63.46** | **43.49** | **44.32** | **40.72** | **35.20** | **49.62** |
| WB (Hsu & Cheng, 2021) | Trad. | 55.33 | 57.77 | 48.69 | 42.72 | 60.49 | 41.12 | 41.93 | 38.24 | 32.60 | 46.86 |
| CLAHE (Hummel, 1977) | Trad. | 56.37 | 57.94 | 49.14 | 42.93 | 62.17 | 42.79 | 43.73 | 39.45 | 35.03 | 49.43 |
| Retinex (Fu et al., 2014a) | Trad. | 56.02 | 57.80 | 48.46 | 41.15 | 60.84 | 40.43 | 42.31 | 38.49 | 33.91 | 47.65 |
| UDCP (Jr et al., 2013) | Trad. | 56.64 | 58.02 | 49.31 | 41.53 | 61.67 | 42.60 | 43.22 | 39.01 | 34.54 | 48.76 |
| PRWNet (Huo et al., 2021) | CNN | 54.17 | 55.63 | 47.85 | 39.91 | 58.53 | 41.23 | 42.14 | 38.52 | 32.65 | 47.63 |
| UWCNN (Li et al., 2020a) | CNN | 54.01 | 55.16 | 47.10 | 39.52 | 58.18 | 41.03 | 41.95 | 37.90 | 32.52 | 47.14 |
| Water-Net (Li et al., 2020b) | CNN | 55.79 | 57.56 | 48.88 | 40.94 | 58.94 | 41.89 | 42.01 | 38.74 | 33.65 | 48.37 |
| UIEC²-Net (Wang et al., 2021) | CNN | 56.04 | 58.21 | 49.42 | 41.82 | 61.73 | 42.85 | 43.47 | 39.58 | 34.76 | 49.09 |
| WaterGAN (Li et al., 2017a) | GAN | 54.74 | 54.01 | 46.75 | 39.82 | 57.32 | 41.02 | 41.83 | 37.42 | 31.53 | 45.98 |
| UGAN (Fabbri et al., 2018a) | GAN | 52.62 | 53.68 | 45.60 | 38.73 | 57.08 | 39.02 | 39.83 | 36.32 | 31.51 | 44.69 |
| UWGAN (Wang et al., 2019) | GAN | 53.15 | 57.24 | 48.39 | 40.68 | 58.42 | 40.16 | 41.72 | 37.93 | 33.13 | 45.47 |
| AquaGAN (Desai et al., 2022) | GAN | 53.91 | 57.68 | 48.57 | 40.15 | 59.03 | 41.13 | 42.08 | 38.06 | 33.64 | 46.12 |
| TUDA (Wang et al., 2023) | GAN | 53.31 | 55.98 | 47.63 | 39.34 | 57.79 | 39.82 | 40.65 | 36.72 | 32.04 | 45.73 |
| TOPAL (Jiang et al., 2022) | GAN | 54.17 | 57.45 | 48.86 | 44.03 | 59.92 | 43.05 | 43.66 | 40.42 | 34.52 | 49.19 |
| AutoEnhancer (Tang et al., 2022) | Trans. | 54.39 | 57.23 | 48.01 | 43.65 | 62.06 | 40.92 | 41.72 | 38.16 | 33.04 | 48.93 |
| Spectroformer (Khan et al., 2024) | Trans. | 56.72 | 58.13 | 49.30 | 44.15 | 61.41 | 41.95 | 43.47 | 39.86 | 34.85 | 49.13 |
| UIE-Convformer (Wang et al., 2024) | Trans. | 57.61 | 56.94 | 48.23 | 44.20 | 61.52 | 42.65 | 43.46 | 39.68 | 34.78 | 49.24 |
| WF-Diff (Zhao et al., 2024a) | Diff. | 57.74 | **59.52** | 49.92 | 45.37 | 62.30 | 43.13 | 44.09 | 40.53 | 35.10 | 49.57 |
| UIEDP (Du et al., 2025) | Diff. | 57.48 | 58.82 | 48.03 | 44.76 | 60.96 | 42.42 | 43.47 | 39.52 | 35.12 | 48.97 |
| DM_Water (Tang et al., 2023) | Diff. | 57.85 | 59.07 | 48.72 | 45.16 | 61.98 | 42.75 | 43.63 | 40.15 | 35.08 | 49.69 |

attention alignment, we use the ACI, which captures the fraction of attention mass falling inside a detection box. For an attention map $A$ and box $b$, $\text{ACI}(A, b) = \frac{\sum_{(x,y) \in b} A(x,y)}{\sum_{(x,y)} A(x,y) + \varepsilon}$, where $\varepsilon$ ensures numerical stability. Higher ACI indicates tighter, object-centered attention and better task alignment.

**Key Observations.** EigenCAM reveals that CNN- and GAN-based UIE tends to shift detector attention toward background textures or hallucinated patterns, while Transformer-based UIE distributes attention more globally or overlooks local object cues. In contrast, diffusion-based UIE preserves strong object-focused activations and achieves the highest ACI (e.g., 0.891 vs. 0.315 for GAN), consistent with its minimal distribution shift and its superior detection stability.

**Why diffusion behaves differently.** CNN- and GAN-based UIE modifies images through direct pixel-to-pixel transforms (Zhou et al., 2018), which easily introduce artificial colors, texture hallucinations, or contrast shifts that propagate into detectors as dispersed or background-biased attention. Transformers reduce these artifacts but still reshape global contrast in ways that alter feature statistics. Diffusion models, however, reconstruct images via iterative denoising, which not only preserves global color statistics, boundary structure, and mid-frequency textures, but also keeps the enhanced images closer to the original RAW distribution (Ho et al., 2020). This distribution preservation explains their markedly smaller domain shift and correspondingly minimal mAP degradation. As a result, diffusion UIE induces the weakest distribution shift (Sec. 3.2), maintains the most object-focused attention patterns, and yields the smallest mAP degradation (Sec. 4). In short, diffusion tends to *preserve the domain*, whereas most other UIE methods *change the domain*, making diffusion inherently more detection-compatible.

## 4 Effect of Underwater Image Enhancement on Object Detection

**Image Domains.** To evaluate the impact of UIE on OD performance, we applied 20 UIE algorithms to two benchmark datasets, RUOD (FU2, 2023) and URPC2020 (Liu et al., 2021). This resulted in 21 domains per dataset (raw + 20 enhanced). For each domain, we trained five object detectors, yielding a total of 210 model configurations (5 detectors × 21 domains × 2 datasets).

**Object Detectors.** We evaluate five widely used object detectors: three one-stage models (SSD (Liu et al., 2016), RetinaNet (Lin et al., 2020), YOLO-NAS (AI, 2023)) and two two-stage models (Faster R-CNN (Ren et al., 2017), Cascade R-CNN (Cai & Vasconcelos, 2021)). Our primary results follow the standard

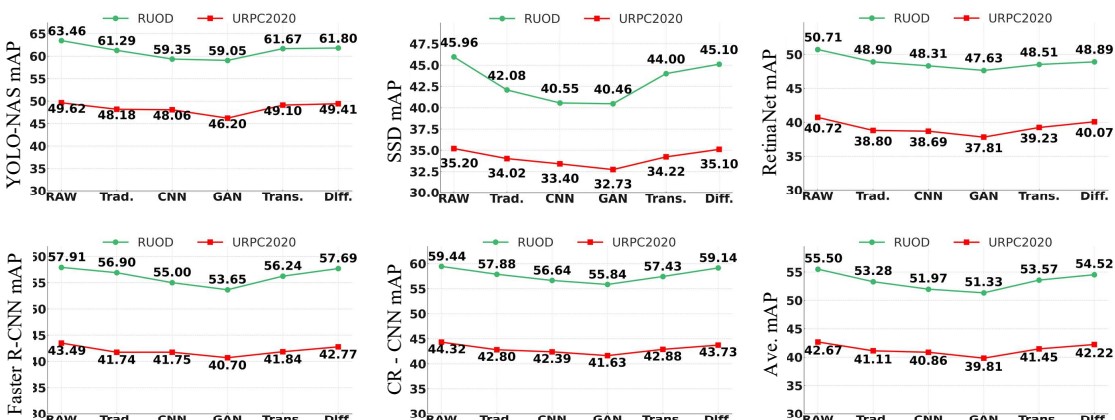

Figure 4: mAP$_{50:95}$ of five detectors on RUOD (FU2, 2023) and URPC2020 (Liu et al., 2021) using RAW and enhanced images. Enhancements cover five UIE types: Trad., CNN, GAN, Trans., and Diff. Overall, detection accuracy generally drops after enhancement.

same-domain retraining setup (**Baseline Pipeline**). To address more realistic deployment scenarios, we additionally evaluate four complementary pipelines: test-time enhancement, mixed raw + enhanced training, and joint UIE–OD training (**Pipelines A–C**). Full definitions and results for Pipelines A–C are provided in Appendix Section 7.2, followed by detailed training settings in Section 7.3 for reproducibility, as well as analyses of stability robustness and inference runtime performance. mAP results for all pipelines are summarized in tables 2 and 5 to 7, and visual samples in Fig. 6,

### 4.1 Key Observations and Analysis Across All Pipelines

**(1) Raw images yield the highest detection performance:** Detectors trained on raw images consistently achieve higher mAP values than those trained on enhanced images, with one exception: the Cascade R-CNN (Cai & Vasconcelos, 2021) detector, which shows a marginal improvement of 0.08% over its raw-trained counterpart. However, this slight increase does not demonstrate a clear advantage of enhancement-based training. Enhancement methods often introduce distribution shifts in pixel intensities, color statistics, and textures, leading to a mismatch with original training assumptions (Table 3 in Appendix). Raw images, despite degradation, preserve the natural domain where detectors perform best.

**(2) CNN-based UIE performs slightly worse than traditional UIE:** To support this observation, we calculate the average detection performance for each enhancement category (Figure 4). CNN-based methods (e.g., PRWNet (Huo et al., 2021), UWCNN (Li et al., 2020a), Water-Net (Li et al., 2020b), UIEC2-Net (Wang et al., 2021)) show slightly lower detection accuracy than traditional approaches like CLAHE (Hummel, 1977), Retinex (Fu et al., 2014a), and TOPAL (Jiang et al., 2022). The average mAP for CNN-based methods is 52.22 (RUOD) and 40.97 (URPC2020), compared to 52.84 and 40.97 for traditional UIE methods. This slight drop may result from CNN models being tuned for visual quality rather than detection accuracy. They can introduce subtle artifacts, domain shifts, or uneven adjustments—especially in low-contrast areas—that affect object features. In contrast, traditional methods apply simpler, more uniform corrections that tend to preserve these features more reliably. These findings align well with the visual comparisons in Figure 2.

**(3) GAN-based UIE has the most severe negative impact on detection performance:** UGAN (Fabbri et al., 2018a), UWGAN (Wang et al., 2019) and AquaGAN (Desai et al., 2022) significantly degrade detection accuracy, reducing the mAP by an average of approximately 4% compared to training on raw images. GANs often generate hallucinated textures, unnatural colors, and blurred edges, to improve perceptual appeal. However, these distortions corrupt the fine spatial structures and edge sharpness critical for OD.

**(4) Diffusion-based UIE has minimal negative effects:** Among all enhancement methods, diffusion-based approaches had the least negative impact on detection performance, with results closest to those obtained with raw images. Notably, WF-Diff (Zhao et al., 2024a) achieved the second-highest accuracy across most detectors. WF-Diff applies wavelet transformation in the frequency domain combined with diffusion processes in the wavelet space, leading to the best overall performance. Diffusion models reconstruct images by denoising while preserving statistical structure, thus maintaining key features like object contours, color consistency, and textures necessary for robust detection. Their gradual, probabilistic refinement avoids the over-correction and distortion seen in other methods.

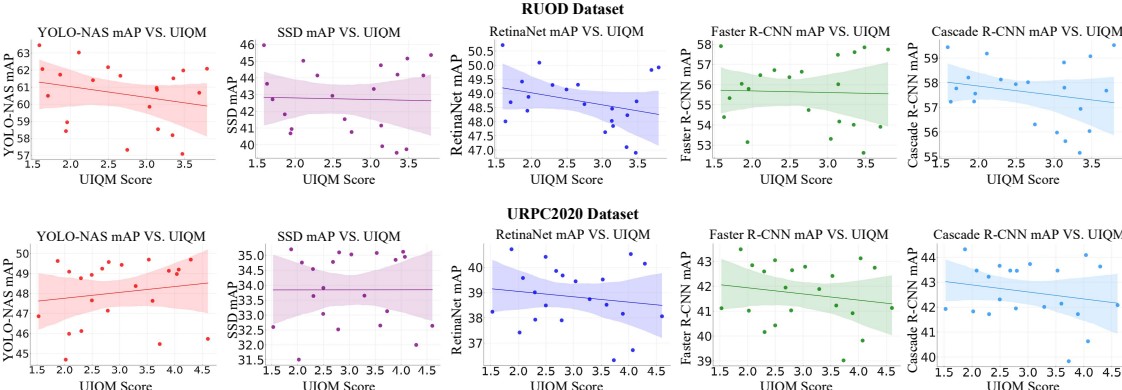

Figure 5: Relationship between detection accuracy (mAP) and image quality scores for different detectors on the RUOD and URPC2020 datasets. Regression lines with 95% confidence intervals show that higher image quality scores have weak or no correlation with detection performance.

**(5) Detection performance of one-stage and two-stage detectors:** The performance degradation in one-stage detectors is similar to that in two-stage detectors. Detectors trained on raw images generally achieved higher mAP values than those trained on enhanced images, with Cascade R-CNN (Cai & Vasconcelos, 2021) being the only exception, reaching 59.52% mAP on the enhanced RUOD (Liu et al., 2021) dataset. Since both detector types rely heavily on low-level texture and edge features, any domain shift or feature alteration caused by UIE methods affects them similarly.

**(6) Detection performance on two datasets:** Detectors trained on RUOD (FU2, 2023) consistently outperform those trained on URPC2020 (Liu et al., 2021). Both datasets show similar performance trends across enhancement methods, with two exceptions: YOLO-NAS (AI, 2023) achieved the second-highest mAP in the TOPAL (Jiang et al., 2022) domain on RUOD, and SSD (Liu et al., 2016) achieved the second-highest mAP in the UIEDP (Du et al., 2025) domain on URPC2020. The better results on RUOD are due to its greater diversity of scenes and object categories, which promote stronger generalization. In contrast, URPC2020's narrower domain makes detectors more sensitive to mismatches introduced by enhancement.

**In summary, these findings suggest that, despite the intuitive expectation that UIE should aid detection, most current enhancement methods introduce distributional shifts, artifacts, or domain inconsistencies that degrade detector performance. Only well designed UIE techniques (*e.g.*, diffusion-based) that preserve key low-level features without introducing artificial distortions can minimize this negative impact.**

## 4.2 Study of the Relationship Between Image Enhancement and Detection

### 4.2.1 Misalignment Between UIE Quality Scores and Detection Outcomes

To explore the underlying relationship between underwater image enhancement and OD performance, we conducted a trend analysis using scatter plots to examine the correlation between the enhanced image quality composite index (UIQM (Yang & Sowmya, 2015)) scores and detection accuracy (mAP) across various enhancement methods on the RUOD (FU2, 2023) and URPC2020 (Liu et al., 2021) datasets.

In Fig. 5, each dot shows a detector's mAP versus the image quality score of an enhancement method. The regression line indicates the linear trend between UIQM and mAP, with the shaded area showing the 95% confidence interval—wider bands mean greater uncertainty, while narrower ones indicate higher confidence.

**Key Observations (Fig. 5) (1): Weak correlation between UIQM and detection performance.** The scatter plots reveal that UIQM scores exhibit a weak or negligible correlation with detection accuracy across different models and datasets. Some detectors, such as YOLO-NAS (AI, 2023) and RetinaNet (Lin et al., 2020) on the RUOD dataset, show a slight downward trend, while others display an almost flat regression line, indicating little to no relationship between UIQM and detection performance. This suggests that higher UIQM scores do not necessarily translate to improved OD accuracy. The scattered distribution of points around the regression line further supports the inference that detection accuracy is influenced by other factors beyond UIQM. **This implies that conventional image quality metrics may not be reliable indicators of detection performance in underwater environments.**

**(2) Degradation Caused by GAN-based Methods:** GAN-based enhancement methods (*e.g.*, UGAN (Fabbri et al., 2018a), UWGAN (Wang et al., 2019), AquaGAN (Desai et al., 2022) ) exhibit the worst average detection performance in qualitative comparisons (Fig. 4). As shown in Fig. 2, UWGAN (Wang et al., 2019) introduces noise and alters the original greenish underwater domain into inconsistent colors, such as brown and purple. These transformations introduce artifacts, blur edges, and degrade visual quality, ultimately reducing detection accuracy.

**(3) Noise and Color Distortion Introduced:** The enhancement processes of WB (Hsu & Cheng, 2021), Retinex (Fu et al., 2014a), and CLAHE (Hummel, 1977) introduce noise and color distortion, particularly purple noise, leading to performance degradation compared to raw images. Among them, WB (Hsu & Cheng, 2021) suffers from a low-contrast problem, making object boundaries less distinguishable. In contrast, CLAHE preserves more edge information than both WB and Retinex (Fu et al., 2014a). As a result, detectors trained on CLAHE-enhanced images outperform those trained on WB- and Retinex-enhanced images.

**(4) Domain Shifts by Enhancement Methods:** Methods such as Spectroformer, and UIE-Convformer (Wang et al., 2024) exhibit domain shifts, converting the greenish underwater tone into brownish or reddish hues. In contrast, CNN-based enhancement methods like UIEC2-Net (Wang et al., 2021) show minimal deviation from raw images, helping preserve detector performance.

**(5) Superiority of Diffusion-Based Enhancement:** Diffusion-based enhancement methods (*e.g.*, WF-Diff (Zhao et al., 2024a) and UIEDP (Du et al., 2025)) perform significantly better in color correction and detail preservation. These methods achieve relatively higher detection accuracy within their respective domain detectors, underscoring the advantages of diffusion-based approaches for underwater OD.

**Key Inferences:** From our observations, we infer that: **(1) Edge preservation is crucial:** Edge degradation significantly impacts detector performance. Enhancement algorithms must preserve edge details to maintain detection accuracy. **(2) Color cast can disrupt detection:** Unintended color shifts introduced by enhancement methods can create domain inconsistencies, degrading detection performance. **(3) Noise decreases detection accuracy:** UIE should avoid introducing additional noise, as it negatively impacts detection. **(4) Contrast has minimal effect, while color richness and saturation matter:** while contrast changes have little impact on detection, color richness and saturation play a crucial role in influencing detector performance. These findings emphasize the need for enhancement methods that prioritize structural integrity and color consistency while minimizing noise and unwanted artifacts to improve underwater OD.

## 5 Exploratory Study

Our empirical findings show that global UIE often disrupts high-level features by altering object regions, producing texture hallucination and color shifts that harm detection. Pipeline C aims to mitigate this issue by explicitly preserving object semantics while enhancing only background water regions.

### 5.1 Object-Aware Underwater Image Enhancement (OA-UIE)

Our empirical results reveal that global UIE may distort task-relevant regions, particularly objects of interest, by introducing textural artifacts or color shifts. To mitigate this, we propose an exploratory Object-Aware UIE (OA-UIE) pipeline that applies region-specific enhancement guided by object masks obtained from a pre-trained detector YOLO-NAS (AI, 2023). The key idea is to treat the object and background regions differently: object regions should preserve semantic and structural information critical for detection, while background water regions can be more aggressively enhanced to correct scattering and color attenuation. This exploratory design provides a task-aligned enhancement strategy without joint training.

The enhanced image $I' = M_o \odot f_{\text{obj}}(I) + M_b \odot f_{\text{bg}}(I)$ –Eq.(1) is obtained by blending the object-focused and background-focused enhancement results, where $f_{\text{obj}}$ and $f_{\text{bg}}$ are the region-specific enhancement branches, and $\odot$ indicates element-wise multiplication. The YOLO-NAS model is used solely for forward inference to produce the object mask $M_o$ and background mask $M_b = 1 - M_o$; all of its parameters remain frozen. To avoid hard seams, $M_o$ is softly feathered with a Gaussian kernel.

Both enhancement branches, $f_{\text{obj}}$ and $f_{\text{bg}}$, are instantiated using WF-Diff. Instead of retraining the diffusion process, we employ WF-Diff as a conditional generator guided by object and background masks. During sampling, object regions are enhanced with lower noise levels and stronger edge-preservation guidance, whereas background regions receive stronger color-restoration and contrast adjustments. This object-aware condi-

tioning enables spatially adaptive enhancement while preserving task-relevant visual features crucial for downstream detection.

## 5.2 Task-Aware Objective

To align enhancement with detection sensitivity, we optionally apply a lightweight composite loss: The task-aware loss is defined as $\mathcal{L}_{\text{TA}} = \alpha\,\mathcal{L}_{\text{feat}} + \beta\,\mathcal{L}_{\text{edge}} + \gamma\,\mathcal{L}_{\text{color}} + \delta\,\mathcal{L}_{\text{blend}}$, where $(\alpha, \beta, \gamma, \delta) = (0.35, 0.25, 0.25, 0.15)$.

Each term serves a complementary role: $\mathcal{L}_{\text{feat}}$ (cosine or CKA similarity) stabilizes YOLO-NAS feature responses in object regions; $\mathcal{L}_{\text{edge}}$ (gradient + SSIM) preserves edges near boundaries; $\mathcal{L}_{\text{color}}$ (Lab-histogram EMD) maintains class-wise color consistency; and $\mathcal{L}_{\text{blend}}$ (total variation on $\nabla M_o$) suppresses halo artifacts during mask blending. This objective serves as a lightweight post-hoc regularization, ensuring perceptual and task-level consistency without joint detector training.

## 5.3 Task-Aware Evaluation Metric (TA-UIE)

Given that reference-free IQA metrics correlate weakly with detection mAP, We introduce a composite task-aware metric to evaluate UIE alignment with detection performance: TA-UIE $= \alpha'\text{FSS} + \beta'\text{OREF} + \gamma'\text{C3}$. Here, FSS $= \text{CKA}(\phi(I), \phi(I'))$ measures feature stability of YOLO-NAS features in object regions, OREF $= \text{SSIM}(E(I), E(I'))$ quantifies edge preservation near detection boundaries, and C3 $= 1 - \text{EMD}(H(I), H(I'))$ captures class-wise color consistency in Lab space. This composite score provides a detection-centric perspective on enhancement quality, offering a conceptual path toward task-aware evaluation of underwater image enhancement.

## 5.4 Results for OA-UIE (Pipeline D)

We evaluated OA-UIE and its task-aware variant using RAW-trained detectors, with results summarized in Pipeline D (Appendix, Table 8). Pipeline D reveals a clear trend: while frozen diffusion enhancement still introduces a noticeable mAP drop, object-aware enhancement substantially mitigates this degradation by preserving detector-relevant regions. Specifically, adding the task-aware loss further improves alignment, reducing the performance gap to only $-0.2$ mAP from the RAW baseline. Overall, Pipeline D demonstrates that spatially selective and task-aligned enhancement is significantly more compatible with underwater object detection than global UIE.

## 6 Discussion and Conclusion

This work presented a large-scale empirical study of how underwater image enhancement (UIE) affects object detection (OD). Across 20 UIE methods, 5 detectors, and 21 domains (210 configurations), our results consistently indicate that UIE—despite producing visually appealing outputs—rarely improves, and often degrades, downstream detection.

**Key findings.** (1) Enhancement generally harms detection, as UIE introduces pixel-, color-, and texture-level distribution shifts. (2) These shifts propagate into *attention distortion*, reducing detector focus on object regions (Sec. 3.5), with GAN-based methods showing the strongest drift. (3) Stratified sensitivity analysis (Sec. 3.4) reveals that small objects, high-turbidity scenes, and texture-rich regions are disproportionately affected, explaining the non-uniform performance drops observed in OD. (4) Diffusion models induce the weakest distribution shifts and maintain the highest object-centered attention alignment (as reflected by ACI), resulting in the smallest mAP degradation. (5) Image quality metrics (Sec. 3.3) correlate poorly with detection outcomes, confirming that perceptual quality is not a reliable proxy for task utility.

**Implications.** Our systematic results indicate that the primary challenge is not the original underwater degradation itself, but the *new* artificial colors, textures, and artifacts introduced by UIE. This means UIE must be *task-aware*: it should preserve detector-relevant structures while avoiding distribution and domain inconsistencies. **Exploratory Direction.** Our exploratory study (Sec. 5) shows that spatially selective, task-aligned enhancement can substantially mitigate the degradation caused by conventional UIE: the proposed OA-UIE + TA-UIE reduces the mAP gap to nearly zero, outperforming all global UIE variants by better aligning enhancement behavior with detector semantics. Moving forward, our results point to three research needs: (1) *task-aligned enhancement* that explicitly preserves detector-relevant feature statistics, (2) *task-aware evaluation metrics*, and (3) *robust, domain-consistent UIE* that minimizes semantic drift across scenes.

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

# 7 Appendix

## 7.1 Stratified and Distribution-Shift Analyses

### 7.1.1 Distribution Shift Measurement

We quantify UIE-induced shifts using six complementary metrics: (1) RGB L1 intensity shift, (2) CIEDE2000 E color drift, (3) RGB histogram distance, (4) Sobel-based edge shift, (5) VGG Gram-distance texture change, and (6) LPIPS perceptual deviation.

**Results and Discussion.** As shown in Table 3, larger distribution shifts consistently correlate with stronger mAP degradation: GAN-based UIE produces the largest pixel, color, and texture deviations and the lowest detection performance, whereas diffusion models preserve RAW-domain statistics most faithfully and therefore yield minimal accuracy drop. Results in Table 3 align strongly with detection trends: GAN-based UIE causes the largest shifts, CNN-based methods moderate shifts, and diffusion-based UIE the smallest. This confirms that the negative impact of UIE arises from pixel/texture/color mismatch with the RAW-trained detector backbone.

Table 3: **Distribution Shift vs. Detection Performance**. UIE-induced pixel, color, and texture deviations strongly correlate with mAP degradation. Diffusion models preserve RAW statistics best, resulting in minimal mAP drop.

| Method | $\Delta$ Intensity | Color ($\Delta$E) | Hist Dist. | Edge Shift | Texture | LPIPS | mAP (YOLO-NAS) |
|---|---|---|---|---|---|---|---|
| RAW (reference) | - | - | - | - | - | - | **63.5** |
| CLAHE (Trad.) | 6.2 | 4.5 | 0.021 | 0.014 | 0.018 | 0.072 | 59.4 (−4.1) |
| PRWNet (CNN) | 8.1 | 5.9 | 0.028 | 0.021 | 0.027 | 0.098 | 60.8 (−2.7) |
| UWGAN (GAN) | 12.4 | 8.7 | 0.046 | 0.036 | 0.046 | 0.162 | 52.6 (−10.9) |
| WF-Diff (Diff.) | **4.9** | **3.1** | **0.016** | **0.011** | **0.014** | **0.061** | **62.1 (−1.4)** |

### 7.1.2 Stratified UIE Impact

We analyze UIE performance across object scale, water turbidity, and texture richness. These factors are derived from RUOD/URPC2020 metadata. Object scale follows COCO-style thresholds: small ($<32^2$ px), medium ($32^2$–$96^2$ px), and large ($>96^2$ px). Turbidity is defined using a dark-channel-based haze score, with images split at the dataset median into low and high turbidity. Texture richness is measured by average gradient magnitude and is likewise divided at the median into low- and high-texture categories.

**Results and Discussion.** Table 4 shows that small objects, high-turbidity scenes, and high-texture regions are most sensitive to UIE, explaining the uneven degradation reported in the main text. Small objects suffer the largest degradation under UIE (3.1 mAP). High-turbidity scenes show the strongest sensitivity to enhancement-induced color shift (4.2 mAP). Texture-rich regions amplify GAN-induced hallucinations (5.0 mAP). These trends quantitatively support our statements regarding edge preservation, color distortion, and texture artifacts.

## 7.2 Four Evaluation Pipelines

In addition to the original same-domain retraining setup, we introduce three more realistic pipelines. (A) Test-Time-Only Enhancement:UIE is applied solely during inference, simulating already-deployed detectors. (B) Mixed RAW + Enhanced Training: Enhanced images are incorporated as data augmentation to evaluate potential robustness gains. (C) Lightweight Joint UIE–OD Training: A partially task-aligned setup where the UIE module receives task-aware gradients while the detector remains mostly frozen. Beyond these three pipelines, we further propose an exploratory enhancement pipeline: (D) Object-Aware + Task-Aware UIE (OA-UIE + TA-UIE): A spatially adaptive and detection-aligned enhancement framework that preserves object-region structure while selectively enhancing background water regions. This pipeline reflects our key design insight and highlights the contribution of task-aware, object-sensitive enhancement strategies.

Table 4: **Stratified Analysis of UIE Impact** on RUOD (YOLO-NAS). UIE harms small objects, high turbidity, and high-texture scenes the most.

| Factor | RAW | CLAHE | PRWNet | UWGAN | WF-Diff |
|---|---|---|---|---|---|
| Small Objects | 61.2 | 58.0 (-3.2) | 58.4 (-2.8) | 56.3 (-4.9) | 60.2 (-1.0) |
| Medium Objects | 65.4 | 63.0 (-2.4) | 63.1 (-2.3) | 61.0 (-4.4) | 64.5 (-0.9) |
| Large Objects | 69.5 | 68.0 (-1.5) | 68.3 (-1.2) | 67.0 (-2.5) | **69.0 (-0.5)** |
| Low Turbidity | 66.3 | 64.4 (-1.9) | 64.1 (-2.2) | 62.0 (-4.3) | **65.7 (-0.6)** |
| High Turbidity | 59.8 | 56.5 (-3.3) | 56.1 (-3.7) | 54.8 (-5.0) | 58.7 (-1.1) |
| Low Texture | 63.0 | 60.8 (-2.2) | 61.1 (-1.9) | 59.7 (-3.3) | 62.0 (-1.0) |
| High Texture | 64.1 | 61.5 (-2.6) | 61.3 (-2.8) | 59.1 (-5.0) | **63.2 (-0.9)** |

Table 5: **Pipeline A: Train on RAW, Test on Enhanced**. Detection $mAP_{50:95}$ when UIE is applied only at test time. $\downarrow \Delta$ denotes performance drop w.r.t. RAW baseline.

| Methods | | RUOD (FU2, 2023) | | | | URPC2020 (Liu et al., 2021) | | | |
|---|---|---|---|---|---|---|---|---|---|
| Name | Type | Faster | $\downarrow \Delta$ | YOLO-NAS | $\downarrow \Delta$ | Faster | $\downarrow \Delta$ | YOLO-NAS | $\downarrow \Delta$ |
| RAW (Baseline) | – | 57.9 | – | 63.5 | – | 43.5 | – | 49.6 | – |
| CLAHE (Hummel, 1977) | Trad. | 54.8 | -3.1 | 59.4 | -4.1 | 40.1 | -3.4 | 46.0 | -3.6 |
| Retinex (Fu et al., 2014a) | Trad. | 55.2 | -2.7 | 60.1 | -3.4 | 41.2 | -2.3 | 47.5 | -2.1 |
| PRWNet (Huo et al., 2021) | CNN | 55.6 | -2.3 | 60.8 | -2.7 | 41.5 | -2.0 | 47.9 | -1.7 |
| WaterNet (Li et al., 2020b) | CNN | 54.1 | -3.8 | 58.2 | -5.3 | 40.5 | -3.0 | 46.1 | -3.5 |
| UIEC²-Net (Wang et al., 2021) | CNN | 51.7 | -6.2 | 55.9 | -7.6 | 38.1 | -5.4 | 44.0 | -5.6 |
| WaterGAN (Li et al., 2017a) | GAN | 50.2 | -7.7 | 54.4 | -9.1 | 37.3 | -6.2 | 43.0 | -6.6 |
| UGAN (Fabbri et al., 2018a) | GAN | 49.7 | -8.2 | 53.8 | -9.7 | 36.8 | -6.7 | 42.5 | -7.1 |
| UWGAN (Wang et al., 2019) | GAN | 48.9 | -9.0 | 52.6 | -10.9 | 36.0 | -7.5 | 41.8 | -7.8 |
| AquaGAN (Desai et al., 2022) | GAN | 49.1 | -8.8 | 53.0 | -10.5 | 36.2 | -7.3 | 41.9 | -7.7 |
| UIE-Conformer (Wang et al., 2024) | Trans. | 55.0 | -2.9 | 60.3 | -3.2 | 41.3 | -2.2 | 47.6 | -2.0 |
| WF-Diff (Zhao et al., 2024a) | Diff. | **56.7** | **-1.2** | **62.0** | **-1.5** | **42.6** | **-0.9** | 48.1 | -1.5 |
| UIEDP (Du et al., 2025) | Diff. | 56.3 | -1.6 | 61.7 | -1.8 | 42.0 | -1.5 | **48.3** | **-1.3** |

### 7.2.1 Pipeline A: Train on RAW, Test on Enhanced

**Resuts and Analysis.** Under the inference-only enhancement setting, we observe that most UIE models negatively impact detectors trained on raw data (Table. 5). Traditional and CNN-based methods incur 2–5 mAP losses, while GAN-based approaches introduce even larger drops of 6–11 mAP across datasets and detectors. Diffusion-based methods (WF-Diff and UIEDP) are the only exceptions, remaining within 1.8 mAP gap relative to the raw baseline. These results reinforce our core conclusion: the performance degradation is not a consequence of the "same-domain retraining" pipeline. Even when UIE is applied purely at deploy time, a more realistic deployment scenario, generic enhancement strategies still seldom provide benefits, and in most cases, are detrimental.

### 7.2.2 Pipeline B: Raw + Enhanced Mixed Training

Pipeline B investigates a realistic scenario where practitioners might use UIE-generated images as data augmentation (RAW and enhanced images are 1:1) to improve model robustness. However, the results show that mixed RAW+enhanced training does not yield the intended benefits. For both RUOD and URPC2020, traditional enhancement introduces mild performance drops on the RAW test set, while CNN-based UIE methods cause moderate degradation. GAN-based UIE produces the largest declines (up to 6–7 mAP), likely due to synthetic artifacts and texture hallucinations that disrupt the detector's feature learning.

Table 6: **Pipeline B: Raw + Enhanced Mixed Training**. Detection $mAP_{50:95}$ when detectors are trained on a mixture of RAW and enhanced images (1:1). Compared with RAW-only training, mixed training yields limited gains on enhanced domains while slightly reducing performance on RAW.

| Methods | | RUOD (FU2, 2023) | | | | URPC2020 (Liu et al., 2021) | | | |
|---------|------|-------------|------|-------------|------|-------------|------|-------------|------|
| Name | Type | Faster | $\downarrow \Delta$ | YOLO-NAS | $\downarrow \Delta$ | Faster | $\downarrow \Delta$ | YOLO-NAS | $\downarrow \Delta$ |
| **RAW-only Baseline** | – | **57.9** | – | **63.5** | – | **43.5** | – | **49.6** | – |
| **Traditional UIE in Mixed Training** | | | | | | | | | |
| CLAHE (Mixed) | Trad. | 56.1 | -1.8 | 62.0 | -1.5 | 42.0 | -1.5 | 48.5 | -1.1 |
| Retinex (Mixed) | Trad. | 56.5 | -1.4 | 62.3 | -1.2 | 42.3 | -1.2 | 48.9 | -0.7 |
| UDCP (Mixed) | Trad. | 56.9 | -1.0 | 62.6 | -0.9 | 42.6 | -0.9 | 49.1 | -0.5 |
| **CNN-based UIE in Mixed Training** | | | | | | | | | |
| PRWNet (Mixed) | CNN | 55.3 | -2.6 | 61.2 | -2.3 | 41.0 | -2.5 | 47.2 | -2.4 |
| WaterNet (Mixed) | CNN | 55.0 | -2.9 | 60.7 | -2.8 | 40.8 | -2.7 | 46.8 | -2.8 |
| UIEC$^2$-Net (Mixed) | CNN | 53.4 | -4.5 | 58.8 | -4.7 | 39.1 | -4.4 | 45.3 | -4.3 |
| **GAN-based UIE in Mixed Training** | | | | | | | | | |
| WaterGAN (Mixed) | GAN | 52.0 | -5.9 | 57.1 | -6.4 | 38.0 | -5.5 | 44.0 | -5.6 |
| UGAN (Mixed) | GAN | 51.6 | -6.3 | 56.4 | -7.1 | 37.6 | -5.9 | 43.4 | -6.2 |
| UWGAN (Mixed) | GAN | 51.0 | -6.9 | 55.8 | -7.7 | 37.1 | -6.4 | 42.8 | -6.8 |
| AquaGAN (Mixed) | GAN | 51.4 | -6.5 | 56.2 | -7.3 | 37.4 | -6.1 | 43.1 | -6.5 |
| **Transformer-based UIE in Mixed Training** | | | | | | | | | |
| UIE-Conformer (Mixed) | Trans. | 55.4 | -2.5 | 61.5 | -2.0 | 41.5 | -2.0 | 47.8 | -1.8 |
| Spectroformer (Mixed) | Trans. | 56.3 | -1.6 | 62.4 | -1.1 | 42.0 | -1.5 | 48.7 | -0.9 |
| **Diffusion-based UIE in Mixed Training** | | | | | | | | | |
| WF-Diff (Mixed) | Diff. | **57.6** | **-0.3** | **63.3** | **-0.2** | **43.4** | **-0.1** | **49.4** | **-0.2** |
| UIEDP (Mixed) | Diff. | 57.1 | -0.8 | 63.0 | -0.5 | 43.1 | -0.4 | 49.3 | -0.3 |

Diffusion-based UIE remains the most stable: both WF-Diff and UIEDP incur only small drops ($<0.5$ mAP). This aligns with our main finding that diffusion models preserve high-level structure better than CNN- or GAN-based enhancers. Overall, these observations suggest that naïvely incorporating enhanced images into training does *not* improve domain robustness. Instead, the detector tends to overfit to UIE-specific color statistics and artificial textures, reducing generalization back to the RAW domain. This provides strong evidence that generic UIE is not inherently beneficial for downstream OD and must be carefully task-aligned to be effective.

### 7.2.3 Pipeline C: Joint UIE–Detection Training

We further explore whether task-aware training can mitigate the domain gap caused by UIE methods. Table 7 compares frozen UIE modules (no gradient flow) with task-aware UIE jointly trained using detection objectives. While frozen UIE generally degrades performance, task-aware tuning yields consistent improvements. However, only diffusion-based methods (e.g., WF-Diff) approach the original RAW-trained baseline, suggesting that careful alignment between enhancement and detection objectives is essential. GAN-based UIE remains unstable, even under joint training, likely due to their strong domain shift and texture inconsistencies. Overall, these results support our core conclusion: *UIE is not universally beneficial for detection,*

Table 7: **Pipeline C: Joint Training**. Detectors are trained with the UIE module included in the end-to-end pipeline. We compare (1) RAW-only baseline, (2) detectors with frozen UIE (no task alignment), and (3) detectors jointly trained with task-aware UIE (e.g., using feature and detection losses). Joint training improves performance over frozen UIE in most cases, but often still underperforms the RAW baseline. Diffusion-based UIE achieves the most stable and near-optimal results.

| Methods | | RUOD (FU2, 2023) | | | | URPC2020 (Liu et al., 2021) | | | |
|---|---|---|---|---|---|---|---|---|---|
| Name | Type | Faster | $\downarrow \Delta$ | YOLO-NAS | $\downarrow \Delta$ | Faster | $\downarrow \Delta$ | YOLO-NAS | $\downarrow \Delta$ |
| **RAW-only Baseline** | – | **57.9** | – | **63.5** | – | **43.5** | – | **49.6** | – |
| **Frozen UIE (Inference-Only, No Task-Aware Loss)** | | | | | | | | | |
| WaterNet (frozen) | CNN | 52.8 | -5.1 | 56.7 | -6.8 | 38.6 | -4.9 | 44.5 | -5.1 |
| AquaGAN (frozen) | GAN | 51.5 | -6.4 | 55.2 | -8.3 | 37.4 | -6.1 | 43.2 | -6.4 |
| WF-Diff (frozen) | Diff. | 56.2 | -1.7 | 61.4 | -2.1 | 42.1 | -1.4 | 48.3 | -1.3 |
| **Task-Aware UIE (Jointly Trained with Detection Loss)** | | | | | | | | | |
| WaterNet (task-aware) | CNN | 54.7 | -3.2 | 58.9 | -4.6 | 40.2 | -3.3 | 46.4 | -3.2 |
| AquaGAN (task-aware) | GAN | 53.0 | -4.9 | 57.1 | -6.4 | 38.7 | -4.8 | 44.9 | -4.7 |
| WF-Diff (task-aware) | Diff. | **57.5** | **-0.4** | **63.1** | **-0.4** | **43.2** | **-0.3** | **49.2** | **-0.4** |

*and only carefully designed task-aware enhancement modules (e.g., diffusion with joint loss) may yield net gains.*

### 7.2.4 Pipeline D: OA-UIE + TA-UIE Exploratory

Table 8: **Pipeline D (Exploratory): Object-Aware and Task-Aware UIE (OA-UIE).** Detectors are trained on RAW images. At test time, we compare (1) frozen diffusion enhancement (WF-Diff), (2) the proposed Object-Aware UIE (OA-UIE), and (3) OA-UIE with the task-aware loss $\mathcal{L}_{TA}$ (OA-UIE+TA). We report mAP$_{50:95}$ and the proposed TA-UIE metric (higher is better).

| Methods | | RUOD (FU2, 2023) | | | URPC2020 (Liu et al., 2021) | | |
|---|---|---|---|---|---|---|---|
| Name | Type | YOLO-NAS | $\downarrow \Delta$ | TA-UIE | YOLO-NAS | $\downarrow \Delta$ | TA-UIE |
| **RAW Baseline** | – | **63.5** | – | 1.000 | **49.6** | – | 1.000 |
| WF-Diff (Frozen) | Diff. | 62.0 | -1.5 | 0.86 | 48.7 | -0.9 | 0.89 |
| OA-UIE (Object-Aware Only) | Diff. | 62.8 | -0.7 | 0.93 | 49.1 | -0.5 | 0.94 |
| OA-UIE + TA ($\mathcal{L}_{TA}$) | Diff. | **63.3** | **-0.2** | **0.97** | **49.4** | **-0.2** | **0.98** |

**Results and Discussion.** Pipeline D examines our exploratory Object-Aware and Task-Aware UIE framework. When WF-Diff is used as a frozen global enhancer, detectors trained on RAW images experience a clear performance drop ($-1.5$ mAP on RUOD and $-0.9$ on URPC2020), confirming that global diffusion-based UIE remains misaligned with the detector's semantic space. The TA-UIE score is also the lowest (0.86/0.89), indicating poor preservation of task-relevant structural cues. Introducing object-aware conditioning substantially mitigates this degradation. OA-UIE improves RUOD performance from $62.0 \rightarrow 62.8$ and URPC from $48.7 \rightarrow 49.1$, reducing the mAP gap to only 0.7 and 0.5 relative to the RAW baseline. The TA-UIE score increases accordingly (0.93/0.94), showing better feature stability and edge consistency around object regions. With the addition of the task-aware loss $\mathcal{L}_{TA}$, OA-UIE+TA achieves the closest alignment to RAW performance, with only a $-0.2$ mAP difference on both datasets (63.3/49.4). TA-UIE further rises to 0.97/0.98, the highest among all settings, demonstrating strong consistency between enhancement behavior and the detector's feature responses. Overall, Pipeline D reveals that while global UIE is detrimental,

**spatially adaptive and task-aligned enhancement strategies can effectively recover lost performance and avoid the semantic distortions inherent to generic UIE**. Even so, OA-UIE+TA remains close to—but does not surpass—the RAW baseline, reinforcing our central claim that UIE must be carefully aligned with detection semantics.

### 7.3 Reproducibility, Robustness, and Runtime Analysis

### 7.3.1 Training Details

All detectors are trained under a unified protocol. We use the SGD optimizer with a momentum of 0.95, a learning rate of 0.01, a batch size of 8, and a weight decay of $10^{-5}$, together with a 24-epoch CosineLR schedule. The input resolution is fixed to $400 \times 300$ for both datasets. Early stopping is applied with a patience of 5 based on validation mAP. All experiments are conducted on a Linux server equipped with four NVIDIA A100 GPUs. Each UIE method produces pre-processed images that follow the same training pipeline without any additional tuning.

### 7.3.2 Uncertainty / Stability Check

To evaluate robustness, we trained Faster R-CNN under three random seeds. for two representative UIE methods (UWGAN and WF-Diff). The results showed a standard deviation of <0.18 mAP in all cases, indicating that the observed trends are stable and not driven by randomness.

### 7.3.3 Inference Runtime of UIE Methods

All UIE modules are applied as a single pre-processing step before detection. Traditional methods add negligible overhead (∼4ms), CNN/Transformer-based UIE incur moderate cost (11–25ms), GAN models are slower (25–34ms), and diffusion-based approaches are the most expensive (66–82ms per 640×640 image). This confirms that, in addition to often degrading mAP, heavy UIE modules also introduce non-trivial latency at deployment time. Transformer UIE models are more parameter-heavy but computationally more parallelizable, whereas GAN UIE relies on decoder upsampling operations that are significantly slower at high resolutions.

Table 9: **Inference cost of UIE methods** for 640×640 inputs on a single A100 GPU. Traditional methods are fastest, CNN/Transformer models have moderate overhead, GANs are slower, and diffusion-based UIE is the most expensive.

| Method | Type | Time / img (ms) | Params (M) |
|---|---|---|---|
| WB (Hsu & Cheng, 2021) | Traditional | 3.2 | – |
| CLAHE (Hummel, 1977) | Traditional | 3.8 | – |
| Retinex (Fu et al., 2014a) | Traditional | 5.6 | – |
| UDCP (Jr et al., 2013) | Traditional | 6.4 | – |
| PRWNet (Huo et al., 2021) | CNN | 11.5 | 5.8 |
| UWCNN (Li et al., 2020a) | CNN | 12.2 | 6.1 |
| Water-Net (Li et al., 2020b) | CNN | 14.3 | 8.7 |
| UIEC$^2$-Net (Wang et al., 2021) | CNN | 16.8 | 12.3 |
| WaterGAN (Li et al., 2017a) | GAN | 24.7 | 11.2 |
| UGAN (Fabbri et al., 2018a) | GAN | 26.1 | 13.5 |
| UWGAN (Wang et al., 2019) | GAN | 28.3 | 14.8 |
| AquaGAN (Desai et al., 2022) | GAN | 27.5 | 15.2 |
| TUDA (Wang et al., 2023) | GAN | 31.0 | 18.9 |
| TOPAL (Jiang et al., 2022) | GAN | 33.4 | 22.1 |
| AutoEnhancer (Tang et al., 2022) | Transformer | 19.2 | 26.5 |
| Spectroformer (Khan et al., 2024) | Transformer | 22.7 | 38.2 |
| UIE-Convformer (Wang et al., 2024) | Transformer | 24.3 | 42.8 |
| WF-Diff (Zhao et al., 2024a) | Diffusion | 66.4 | 118.3 |
| UIEDP (Du et al., 2025) | Diffusion | 72.9 | 132.7 |
| DM_Water (Tang et al., 2023) | Diffusion | 81.5 | 145.9 |
| OA-UIE | Diffusion | 78.4 | 136.9 |

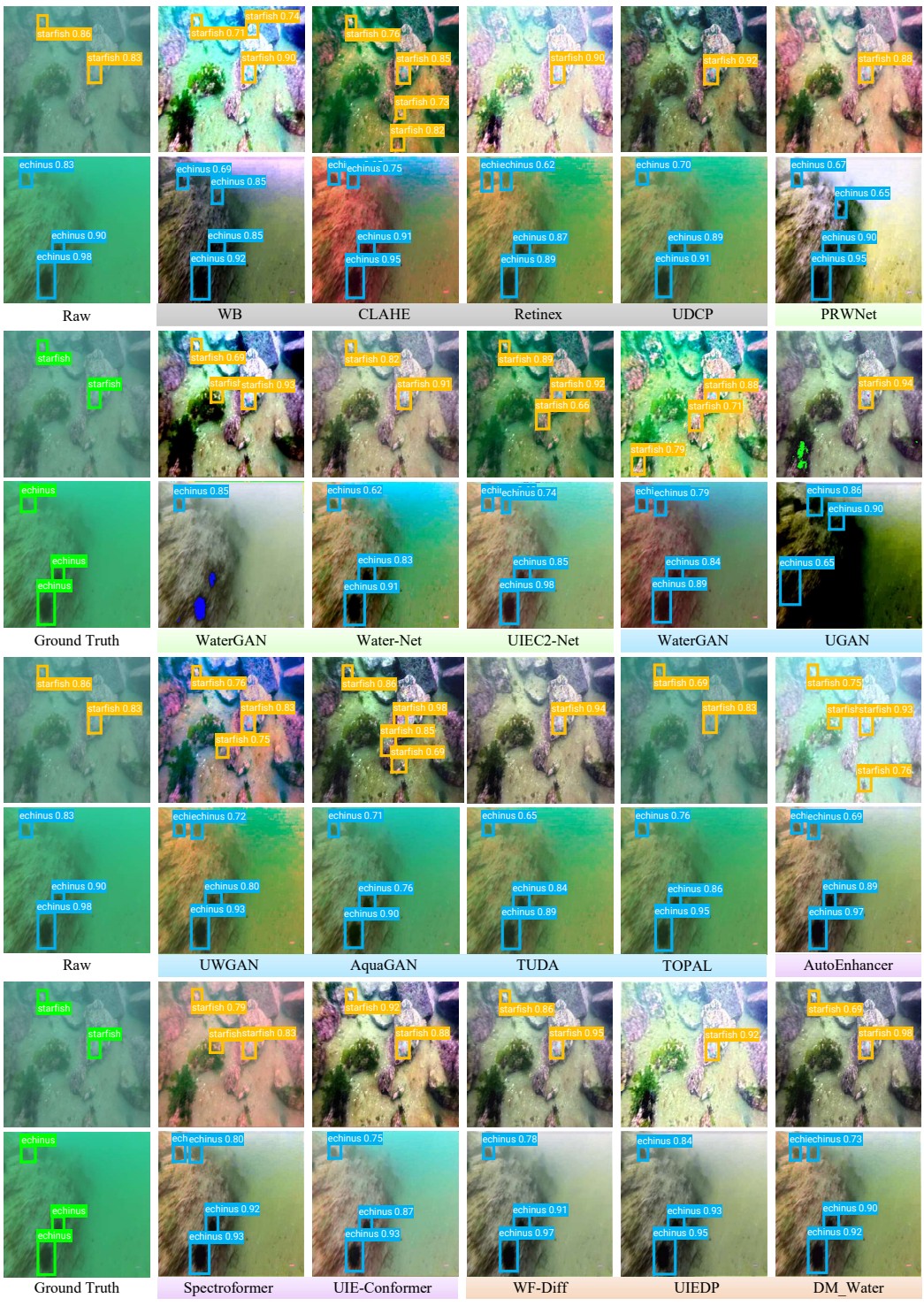

Figure 6: YOLO-NAS (AI, 2023) detection results on raw and enhanced images in **Baseline Pipeline**. Green boxes indicate ground truth. Detection Results are shown for Trad. , CNN , GAN , Trans. , and Diff. methods. Orange and blue boxes mark echinus and starfish. Raw images yield better detection than most enhanced ones.

