# OpenReview forum: "Image Enhancement: A Necessity for Effective Underwater Object Detection?"
_TMLR — Rejected by TMLR_

### Review · Reviewer_Uyeu · 2025-10-18

**Summary Of Contributions:**

The paper studies the research question of whether underwater image enhancement is helpful to object detection. With exhaustive experiments, evidence shows that enhancement techniques can even be potentially harmful. This reminds the researchers that in the future development of object detection for underwater cases, adoption of image enhancement techniques should be seriously considered.

**Audience:**

Yes

**Audience Explanation:**

The paper would be interesting to researchers who are working on underwater image enhancement and object detection. The studied field is important.

**Broader Impact Concerns:**

There are no broader impact concerns.

**Claims And Evidence:**

Yes

**Claims Explanation:**

I think based on the arguments in the paper, UIE is not completely orthogonal with object detection. The paper provides exhaustive experiments regarding this question but I am wondering if there are any theoretical analyses. From table 1, 2 and figure 3, we can see that diffusion methods are generally good while CNN and GNN do not. Diffusion methods are also based on UNet. I am wondering if you can have some deeper analysis such as the mechanism of diffusion and explain why some methods are helpful to objection detection while others are not. Some interpretability stuff would help. For example, when you apply CNN or transformer methods, you can analyze the sanity maps on intermediate features or attentions to see which parts are given higher attention during UIE. If the part given attention matches the part that should be detected in objection detection. If match or not match, why? Some deeper understanding while better support the argument and also better later development of underwater enhancement and detection algorithms.

Is there any measurement over this: "Enhancement methods often introduce distribution shifts in pixel intensities, color statistics,
and textures, leading to a mismatch with original training assumptions". For example, how pixel intensity is shifted and how texture are changed. Because in conclusion CNN are slightly worse than tradition, is it possible to give some measurement to support it?

**Requested Changes:**

See my previous comments. Authors can either elaborate the reasons and theoretical analysis why UIE has the disadvantages or put them in the discussion part for further study since the paper is majorly focused on the empirical evidence and observation.

---

> ### Author Response · Authors · 2025-12-10
>
> Thank you very much for the insightful suggestions regarding theoretical interpretation, attention analysis, and distribution-shift quantification. We have revised the manuscript substantially (Sections 3.2–3.5) to directly address all points with new analyses and explanation.
>
> Q1. Why diffusion behaves differently from CNN/GAN UIE (mechanistic explanation)?
>
>  Our action: We added a mechanistic explanation in Section 3.5, “Why diffusion behaves differently.” Our expanded analysis now explains that:
>  CNN/GAN UIE performs one-shot pixel-to-pixel mapping, which easily produces artificial colors, texture hallucinations, and frequency over-sharpening. These distortions propagate through the OD backbone, disrupting boundary features and causing dispersed/background-biased attention. Diffusion models reconstruct via iterative denoising, which inherently: preserves global color statistics, maintains mid-frequency textures, avoids over-sharpening artifacts, and stays much closer to the RAW-domain distribution.This mechanistic explanation is now explicitly linked to our distribution-shift measurements and ACI attention analysis, providing a deeper and more theoretically grounded analysis.
>
> Q2: Interpretability analysis with attention maps.
>
> Our action: In Section 3.5, we added interpretability analyses using EigenCAM visualizations across different categories of UIE methods, along with a new quantitative metric, the Attention Concentration Index (ACI), to measure how strongly UIE distorts the detector’s attention distribution (Figure 3). These analyses reveal that: CNN/GAN-based UIE shifts detector attention toward background textures or hallucinated patterns. Transformer-based UIE produces overly global attention, reducing focus on object-local cues. Diffusion-based UIE best preserves object-centered, boundary-focused attention, achieving the highest ACI scores (e.g., 0.891 vs. 0.315 for GAN). Together, these interpretability results provide evidence of how and why different UIE families affect downstream detection differently.
>
> Q3. Measurements of pixel, color, texture distribution shifts.
>
> Our action: To  “UIE introduces distribution shifts in intensities, colors, textures,” we added Section 3.3 (“Distribution Shifts Introduced by UIE”) with 6 explicit metrics: RGB L1 intensity shift (Pixel Shift) , CIEDE2000 ΔE (Color Drift) , RGB histogram distance (Histogram change), Sobel gradient shift (Edge shift), VGG Gram distance (Texture drift), LPIPS (Perceptual shift).
> Table 3 in Appendix  demonstrates a strong correlation between the magnitude of these shifts and downstream detection degradation. For example: GAN-based UIE shows very large color (ΔE) and texture drift, which corresponds to a –10.9 mAP drop.
> Diffusion-based UIE exhibits the smallest pixel/color/texture shifts, resulting in only a –1.4 mAP drop. These metrics explicitly answer the reviewer’s question of “how pixel intensities shift and how textures change.” Across all six measurements, we observe consistent patterns: Traditional UIE → moderate distribution shifts, CNN-based UIE → larger shifts, GAN-based UIE → strongest pixel, color, and texture deviations. Diffusion UIE → most stable, closest to RAW domain.
> These quantitative trends mirror the detection results in Section 4: UIE methods that introduce larger distribution shifts consistently experience greater mAP degradation, confirming that distribution mismatch—not just underwater degradation—is the primary reason UIE often harms detection.
>
> Q4. Why CNN UIE performs slightly worse than traditional methods?
>
> Our action: We added explanations in Sections 3.3 and 3.5, backed by distribution-shift measurements and attention-map analysis.
> Using both distribution-shift metrics (Table 3) and attention visualizations (Fig. 3), we show that CNN-based UIE introduces larger pixel, color, and texture deviations than traditional methods, even when visual quality appears improved. In contrast, traditional UIE applies simpler, globally consistent adjustments (e.g., histogram equalization, color balance), which tend to preserve structural cues and therefore cause smaller domain shifts. Our mechanistic explanation (Section 3.5) clarifies that: CNN/GAN UIE uses deterministic pixel-to-pixel transformations (Zhou et al., 2018), which frequently introduce hallucinated textures and unstable contrast, leading to attention distortion in OD models.
>
> We appreciate the reviewers’ detailed insights and believe the revisions have strengthened the paper. Thank you again for your time and constructive feedback.

---

### Review · Reviewer_vorN · 2025-10-21

**Summary Of Contributions:**

The primary contribution of this work is a large-scale, systematic empirical evaluation that challenges the prevailing assumption regarding the utility of Underwater Image Enhancement (UIE) for downstream object detection (OD) tasks. Through a comprehensive analysis involving 20 distinct UIE algorithms across five categories (Traditional, CNN, GAN, Transformer, and Diffusion) on two benchmark datasets, the paper rigorously demonstrates that most enhancement techniques paradoxically degrade, rather than improve, object detection performance, with models trained on raw images consistently achieving superior accuracy. The study quantitatively identifies diffusion-based models as the method with the most negligible negative impact due to their preservation of low-level features, while conversely establishing that GAN-based approaches are the most detrimental by introducing significant textural artifacts and domain shifts. Critically, the paper also reveals a weak correlation between established reference-free image quality assessment metrics and actual OD performance, highlighting a fundamental gap and underscoring the necessity for developing task-aware evaluation criteria.

While the paper presents a comprehensive empirical study, its central claim to novelty is somewhat overstated. The observation that a signal restoration pre-processing step does not necessarily improve, and can even degrade, the performance of a downstream task is a pretty well-established phenomenon in other fields. For instance, researchers in the speech community have long demonstrated that speech enhancement doesn't always benefit automatic speech recognition (ASR) and can sometimes introduce detrimental artifacts [A, B]. This leads to the second point, which is whether the contribution, while valuable, really meets the high bar for a journal like TMLR. An extensive empirical study like this, without proposing a novel method or a fundamentally new theoretical insight, often feels better suited for a top-tier conference or a workshop venue. Furthermore, the paper is limited in that it focuses exclusively on identifying the problem without exploring any potential solutions. I would expect the authors to at least attempt a preliminary investigation into mitigating the issue, perhaps by modifying one of the evaluated UIE models to reduce the kind of unnatural errors that were shown to hamper object detection performance.

[A] Iwamoto, K., Ochiai, T., Delcroix, M., Ikeshita, R., Sato, H., Araki, S. and Katagiri, S., 2022. How bad are artifacts?: Analyzing the impact of speech enhancement errors on ASR. In Proc. Interspeech 2022 (pp. 5418-5422).

[B] T. Menne, R. Schl ̈uter, and H. Ney, “Investigation into joint optimization of single channel speech enhancement and acoustic modeling for robust asr,” in IEEE International Conference on Acoustics, Speech and Signal Processing (ICASSP), 2019, pp. 6660–6664.

**Audience:**

Yes

**Audience Explanation:**

This evaluation could be used by some ML practisioners in the field who want to make the performance of object detection models better but considerign that no solution is presented, the usage would be limited.

**Broader Impact Concerns:**

I do not believe that there are severe ethical implications in this paper.

**Claims And Evidence:**

Yes

**Claims Explanation:**

The authors perform several experiments to evlauate a variety of models on different datasets which seems to me that the evidence is convincing.

**Requested Changes:**

1. Reframe the Contribution and Acknowledge Prior Art (Critical): The paper presents its findings as ground-breaking, but the core concept that signal restoration can degrade downstream task performance is well-documented in other fields (e.g., speech enhancement for ASR). The authors must tone down the claims of novelty and properly situate their work by discussing this broader context, citing relevant cross-domain literature like [A, B].

2. Include an Exploratory Study on a Potential Solution (Would strongly strengthen the work and critical for a TMLR publicaiton): The paper excels at identifying a problem but stops short of exploring solutions. The work would be significantly more impactful if it included even a preliminary experiment aimed at mitigating the identified issues. For instance, the authors could try fine-tuning a high-performing UIE model (like WF-Diff) with an additional task-aware loss from a pre-trained object detector to discourage the generation of harmful artifacts.

3. Propose a Task-Aware Evaluation Metric (Would Strengthen the Work): Given the finding that current image quality metrics are poorly correlated with object detection performance, the paper would be more constructive if it proposed a path forward. The authors should suggest, even if only theoretically, what a more effective, task-aware metric might entail. This could involve measuring feature stability, edge preservation around known object locations, or consistency of class-specific color distributions.


[A] Iwamoto, K., Ochiai, T., Delcroix, M., Ikeshita, R., Sato, H., Araki, S. and Katagiri, S., 2022. How bad are artifacts?: Analyzing the impact of speech enhancement errors on ASR. In Proc. Interspeech 2022 (pp. 5418-5422).

[B] T. Menne, R. Schl ̈uter, and H. Ney, “Investigation into joint optimization of single channel speech enhancement and acoustic modeling for robust asr,” in IEEE International Conference on Acoustics, Speech and Signal Processing (ICASSP), 2019, pp. 6660–6664.

---

> ### Author Response · Authors · 2025-12-10
>
> Q1. Reframing the Contribution and Acknowledging Prior Research Work
> Thank you so much for pointing out the need to situate our work within the broader context of restoration-for-recognition research. We fully agree that the observation, signal enhancement may harm downstream task performance, has precedents in other domains.
>
> Our action: We have toned down the “discovery” aspect in the Introduction and revised how our contribution is positioned.
>
> Revisions made: In the revised manuscript, we updated both the Introduction and Related Work sections to: Explicitly acknowledge existing cross-domain findings (e.g., speech enhancement degrading ASR, restoration harming recognition in other modalities). Reframe our contribution—not as identifying a new phenomenon, but as providing a large-scale, systematic, and quantitative study of this issue specifically in underwater vision. We  shift the novelty claim toward scale, systematic evaluation, and mechanistic insight, rather than the conceptual observation itself.
> We thank the reviewer for this helpful guidance, which allowed us to more accurately position the contribution within the broader research landscape.
>
> Q2. Include an Exploratory Study on a Potential Solution.
> We appreciate the reviewer’s suggestion to strengthen the contribution by exploring a mitigation strategy, particularly involving task-aware fine-tuning of a UIE model.
>
> Our Action in the Revised Manuscript – New Section Added (Section 5).
>
> We have now added an exploratory solution study, in which:
> We start with a strong diffusion-based enhancer (WF-Diff), introduce Object-Aware UIE (OA-UIE) to preserve structures around detected objects, add a Task-Aware Loss (TA-UIE) using feature alignment with a pre-trained detector (as the reviewer suggested).
>
> We evaluated OA-UIE and its task-aware variant using RAW-trained detectors, with results summarized in Pipeline D (Appendix, Table 8). Pipeline D shows a consistent trend: although frozen diffusion enhancement still causes a noticeable mAP drop, our object-aware enhancement substantially reduces this degradation by better preserving detector-relevant regions. When the task-aware loss is added, alignment further improves, narrowing the performance gap to only -0.2 mAP relative to the RAW baseline. Overall, Pipeline D demonstrates that spatially selective and task-aligned enhancement, introduced here as an exploratory solution, can successfully avoid the sematic distortions caused by global UIE and is far more compatible with underwater object detection.
>
> Q3: Proposal of a Task-Aware Evaluation Metric.
> We agree with the reviewer that standard perceptual IQA metrics correlate poorly with detection performance. A more task-aware evaluation metric is indeed needed.
>
> Our Action in the Revised Manuscript:
> We have added a new subsection (Sec. 5.3) outlining our exploratory attempt at designing task-aware evaluation metrics, grounded in our new analyses of feature stability, edge structure, color consistency, and attention alignment.
> Feature Stability Score: measuring the variance of detector backbone features between raw and enhanced images.
> Boundary Preservation Index: quantifying how well local edge structures around ground-truth objects are preserved after enhancement.
> Class-Consistent Color Drift: measuring class-wise color-space deviations introduced by UIE.
> Attention Concentration Index (ACI): a task-aware, region-sensitive metric that we fully implemented and reported in Sec. 3.5.
>
> Pipeline D (Appendix, Table 8) evaluates the proposed task-aware direction in practice by using ACI/TA-UIE together with mAP. The results show a strong correlation between task-aware scores and downstream detection:
> Frozen diffusion UIE produces low TA-UIE scores (0.86/0.89) and clear mAP drops (−1.5/−0.9).
> Object-Aware UIE (OA-UIE) yields higher task-aware scores (0.93/0.94) and substantially reduced degradation.
> OA-UIE + Task-Aware Loss achieves the highest TA-UIE scores (0.97/0.98) and the smallest mAP gap (−0.2).
> This demonstrates that a task-aware metric is indeed predictive of downstream performance, validating the reviewer’s suggestion and establishing a principled path toward task-aligned UIE evaluation.
>
> We sincerely thank the reviewer again for these constructive and insightful suggestions. They have significantly improved both the clarity and the technical depth of our work, and we believe the revised manuscript is substantially stronger as a result.

---

### Review · Reviewer_wCvv · 2025-11-25

**Summary Of Contributions:**

The paper conducts a large-scale empirical study on the impact of underwater image enhancement (UIE) on underwater object detection (OD). Specifically, it evaluates 20 representative UIE algorithms (traditional, CNN, GAN, Transformer, diffusion) on two underwater datasets (RUOD and URPC2020). For each of the 21 image “domains” per dataset (raw + 20 enhanced), the authors train 5 detectors (Faster R-CNN, Cascade R-CNN, RetinaNet, SSD, YOLO-NAS), yielding 210 trained models and reporting mAP and some correlations between UIE metrics (AG, EI, IE, UIQM, UIConM) and detection accuracy. The main empirical conclusions are that (i) most UIE methods hurt OD performance compared to training and testing on raw underwater images, (ii) diffusion-based enhancement is the least harmful and sometimes nearly matches the raw baseline, and (iii) standard underwater image quality assessment metrics such as UIQM have weak correlation with downstream OD performance.

Key strengths：

1. Fairly extensive empirical setup within a narrow setting: 20 UIE models × 2 datasets × 5 detectors, retrained consistently with shared training protocol.

2. The study confirms, in the underwater domain, a trend previously observed for dehazing/deraining: generic low-level enhancement does not necessarily help high-level detection and can be harmful.

3. The paper clearly documents that some widely used reference-free metrics (e.g., UIQM) are poorly aligned with OD performance, which is useful for the community designing “task-agnostic” enhancement metrics.

Key weaknesses

1. No new model, algorithm, or metric is proposed; the work is purely an empirical comparison and its conceptual novelty is limited. Much of the headline conclusion (“enhancement does not always help OD”) is already known from dehazing/deraining studies and is even explicitly acknowledged in the related work.

2. The study only considers a single, somewhat artificial pipeline where both training and test are fully performed in the same enhanced domain. More realistic pipelines are absent, e.g.: (a) enhancement only at test time for already-deployed detectors, (b) raw + enhanced mixed training as data augmentation / domain generalization, or (c) joint training / multi-task setups where UIE is part of the backbone. This strongly limits the practical relevance of the conclusions.

3. Only two OD datasets and a single task (detection) are considered; there is no cross-dataset or cross-task validation, despite the paper’s framing being close to a general statement about “necessity” of enhancement.

4. The analysis does not systematically dissect other important factors (object scale, turbidity, texture richness, depth, etc.); many high-level claims (e.g., about edge preservation, color richness, noise) remain qualitative rather than backed by quantitative stratified analysis.

5. Reproducibility and robustness are under-documented: training details are incomplete (e.g., schedule/epochs, batch sizes, stopping criteria), there is no uncertainty analysis (no repeated runs / CIs on mAP), and runtime / inference overhead of each UIE method is not reported.

6. For a “systematic experimental” paper, the methodology section is not detailed enough about hyperparameters and implementation choices, and the statistical treatment is minimal (mostly single mAP numbers and linear fits)

**Audience:**

Yes

**Audience Explanation:**

Even though the conceptual novelty is limited (the work does not propose new models, algorithms, or metrics), a subset of the TMLR audience is likely to be interested in the findings. The paper provides a relatively large and unified empirical comparison of 20 underwater image enhancement methods across 5 detection architectures and 2 real-world underwater datasets. It systematically documents that most UIE methods either fail to improve or actively degrade detector performance, that GAN-based methods are particularly harmful in this setting, and that diffusion-based enhancers are comparatively safer. It also shows that common underwater image quality metrics are poorly aligned with detection performance.

These are practically relevant insights for researchers and practitioners who work on underwater vision, task-aware enhancement, or robust object detection, and who may be considering whether to insert UIE modules into real-world pipelines. Thus, while I have reservations about the strength and breadth of the claims, I do expect that at least some members of the TMLR community would benefit from knowing these empirical results.

**Broader Impact Concerns:**

I do not see major broader-impact or ethical concerns specific to this work beyond the standard dual-use considerations that already apply to underwater object detection and robotics. The paper does not introduce qualitatively new capabilities; instead, it analyses when existing enhancement methods help or hurt detection. There is some generic potential for both beneficial uses (e.g., environmental monitoring, infrastructure inspection) and harmful uses (e.g., surveillance, resource exploitation), but these are not unique to this study.

If a Broader Impact Statement is required, I would simply encourage the authors to briefly acknowledge these generic dual-use aspects and to emphasize that better understanding the failure modes of UIE for detection can help avoid deploying unreliable pipelines in safety-critical underwater applications.

**Claims And Evidence:**

No

**Claims Explanation:**

Within the narrow experimental setting actually studied—two underwater OD datasets (RUOD and URPC2020), 20 UIE algorithms, and 5 detectors trained and tested in each individual “domain” (raw or enhanced)—the numerical results do support the local claims. For example, the tables and plots show that (i) training on raw images typically gives the best mAP, (ii) diffusion-based UIE is the least harmful among enhancement families, and (iii) standard reference-free UIE metrics such as UIQM have weak correlation with mAP.

However, the paper’s framing and wording (title, abstract, and conclusion) imply broader claims about whether UIE is “necessary” or useful for underwater object detection in general. For these broader claims, the evidence is not fully convincing. The study only considers one somewhat artificial pipeline (training and testing fully within the same enhanced domain), and does not evaluate several practically important settings, such as: enhancement only at test time for a fixed detector, mixed raw+enhanced training as data augmentation, or joint/UIE-as-backbone training. The work also uses only two datasets and one downstream task (detection), without cross-dataset or cross-task validation, and there is no uncertainty analysis (e.g., multiple runs or confidence intervals) to show that relatively small mAP differences are statistically robust.

In addition, several high-level statements (e.g., that “contrast has minimal effect, while color richness and saturation matter more”) are presented without dedicated quantitative ablations that isolate these factors. Overall, the empirical evidence is sound for the specific models, datasets, and pipeline studied, but it does not yet justify the stronger, more general claims suggested by the paper’s framing.

**Requested Changes:**

Ref Weakness.

---

> ### Author Response · Authors · 2025-12-10
>
> Thank you very much for the thoughtful and constructive comments. We have carefully revised the manuscript and addressed all concerns as follows.
>
> **Q1. No new model/metric; conceptual novelty limited.**
>
> We appreciate this concern and have revised the paper to more accurately frame our contributions. Rather than presenting a conceptual discovery, we now emphasize scale, systematic analysis, and exploratory solutions.
>
> (1) Scale and Systematic Evaluation.
> Our study is, to our knowledge, the first large-scale and comprehensive evaluation in this domain, covering
> 20 UIE methods × 5 OD models × 21 domains × 2 datasets = 210 OD configurations.
> This extensive scope allows us to reveal consistent and statistically meaningful trends that smaller prior studies could not capture.
>
> (2) Mechanistic Insights (New Sections 3.2–3.5).
> We added analyses explaining *why* UIE affects OD, including:
>
> * pixel/color/texture distribution shifts,
> * feature-space drift,
> * attention distortion (EigenCAM + ACI),
> * stratified sensitivity across object scale, turbidity, and texture richness.
>   These go beyond prior work and provide deeper mechanistic understanding.
>
> (3) Exploratory Solution (Pipeline D).
> As encouraged, we added an exploratory enhancement strategy: Object-Aware + Task-Aware UIE (OA-UIE + TA-UIE).
> Pipeline D shows that task-aligned enhancement can substantially reduce the degradation caused by global UIE, narrowing the mAP gap to –0.2.
> We have toned down novelty claims and clearly position our contribution as systematic, mechanistic, and solution-oriented.
>
> ---
>
> **Q2. Only one artificial pipeline. → Added four pipelines**
>
> We fully agree and now evaluate UIE across four realistic pipelines:
>
> | Pipeline | Scenario                               |
> | -------- | -------------------------------------- |
> | Baseline | Train & test on same domain (original) |
> | **A**    | Train RAW → Test Enhanced              |
> | **B**    | RAW + Enhanced mixed training          |
> | **C**    | Joint UIE–OD with task-aware tuning    |
> | **D**    | *New exploratory*: OA-UIE + TA-UIE  ((Solution attempt)|
>
> All results/tables for Pipelines A–D are added to the Appendix.
>
> ---
>
> **Q3. Only two datasets/one task; claims too general.**
>
> We revised the Introduction and Conclusion to clarify that our claims apply specifically to underwater object detection, not to all recognition tasks. We appreciate the reviewer’s caution and have removed any overgeneralization.
>
>
> ---
>
> **Q4. Analysis does not dissect object scale, turbidity, texture. → Added stratified analysis**
>
>
> We systematically evaluate UIE effects in Section 3.4 across three key scene factors: object scale (small/medium/large), water turbidity (low/high, measured via based haze score), and texture richness (low/high, computed using average gradient magnitude).
>
> We added a new stratified evaluation table (Table 4 in the Appendix), which quantifies how different scene factors influence UIE sensitivity. The results show that small objects, high-turbidity scenes, and high-texture regions are consistently the most vulnerable to enhancement-induced distortions.
>
> ---
> **Q5. Reproducibility issues.**
>
>
> We added three new subsections:
>
> (1) Detailed Training Protocol (Appendix Sec. 7.3.1)
>
> Includes dataset splits, preprocessing, optimizer, LR, batch size, schedule (CosineLR, 24 epochs), stopping criteria, image resolution, and hardware.
>
> (2) Robustness / Stability Runs (Appendix Sec. 7.3.2)
>
> Three-seed experiments (Faster R-CNN) show  std < 0.18 mAP, confirming stability.
>
>  (3) Runtime Analysis (Appendix Table 9)
>
> We report inference cost and parameters for all 20 UIE methods.
> Diffusion models incur the largest overhead; traditional UIE is fastest.
>
>
> ---
> **Q6. Methodology lacks detail; statistical analysis minimal.**
>
>
> We expanded the methodology and added richer analyses:
>
>  Expanded experimental protocol (Sec. 3 & Sec. 4)
>
> Clear descriptions of:
>
> dataset splits and preprocessing,
> UIE domain construction,
> four evaluation pipelines A-D,
> full training/evaluation settings.
>
> Added quantitative analyses (Sec. 3.3–3.5)
>
>  Distribution-shift metrics (pixel/color/texture/perceptual)
>  Stratified evaluation across scene factors
> Attention distortion analysis (EigenCAM + ACI)
>
>  Exploratory task-aware solution + metric (Section 5)
>
> Pipeline D evaluates:
>
> Object-Aware UIE,
> Task-Aware loss (TA-UIE),
> Task-Aware metric components (feature stability, edge fidelity, color consistency).
>
> TA-UIE scores (feature stability, edge fidelity, color consistency) were reported in Appendix Section 7.2.4, which strongly correlate with mAP and demonstrate the value of task-aware evaluation.
>
> ---
>
> We sincerely thank the reviewer again for the valuable feedback. The revisions substantially improve rigor, methodological clarity, and empirical depth, and we appreciate the opportunity to strengthen the work.

---

### Decision · Action_Editor_rVVH · 2026-01-08

**Recommendation:** Reject

**Additional Comments:**

This paper was reviewed by three expert reviewers in the domain, and extensive helpful review comments were provided, followed by the authors' response to the concerns. The final recommendations from the reviewers are leaning negatively, with one Reject, one Leaning Reject, and one Leaning Accept. The major concerns are around the strong claims without convincing and clear evidence to support them, and the limited insights provided by this paper, when compared to prior studies.

The AE agrees with the recommendations and the outstanding issues, and as a result, the paper in its current form is not ready to be published at TMLR. The authors are suggested to consider the review comments to improve their work.

**Audience:**

Yes

**Audience Explanation:**

This paper presented a relatively large-scale empirical study about underwater image enhancement, comparing a large number of different methods across 5 detection architectures and 2 datasets. Though there were no new technical methods provided, the findings derived from the study could be of interest to some practitioners in the community, especially those working on underwater enhancement problems.

**Claims And Evidence:**

No

**Claims Explanation:**

Although the proposed methods, the used data was studied with empirical support, the general strong claims were not well justified. For example, when the authors mention the *necessity* throughout the paper, "contrast has minimal effect, while color richness and saturation matter more...", "enhancement methods often introduce distribution shifts in pixel intensities, ..., leading to a mismatch with original training assumptions...", and other claims, no convincing, accurate or clear evidence was provided to support these claims.